# Coalescent RNA-localizing and transcriptional activities of SAM68 modulate adhesion and subendothelial basement membrane assembly

**Zeinab Rekad, Michaël Ruff, Agata Radwanska, Dominique Grall, Delphine Ciais\*, Ellen Van Obberghen-Schilling\***

Université Côte d'Azur, CNRS, INSERM, iBV, Nice, France

**Abstract** Endothelial cell interactions with their extracellular matrix are essential for vascular homeostasis and expansion. Large-scale proteomic analyses aimed at identifying components of integrin adhesion complexes have revealed the presence of several RNA binding proteins (RBPs) of which the functions at these sites remain poorly understood. Here, we explored the role of the RBP SAM68 (Src associated in mitosis, of 68 kDa) in endothelial cells. We found that SAM68 is transiently localized at the edge of spreading cells where it participates in membrane protrusive activity and the conversion of nascent adhesions to mechanically loaded focal adhesions by modulation of integrin signaling and local delivery of β-actin mRNA. Furthermore, SAM68 depletion impacts cell-matrix interactions and motility through induction of key matrix genes involved in vascular matrix assembly. In a 3D environment SAM68-dependent functions in both tip and stalk cells contribute to the process of sprouting angiogenesis. Altogether, our results identify the RBP SAM68 as a novel actor in the dynamic regulation of blood vessel networks.

## Editor's evaluation

This paper provides important evidence that the RNA binding protein SAM68 regulates endothelial cell migration through multiple mechanisms including localizing actin mRNA to focal adhesions and stimulating transcription of the fibronectin gene. The evidence is generally convincing, although the relative roles of transcription and RNA localization in SAM68 functions and the dynamics of RNA movement to adhesion sites remain unknown. The paper will be of interest to cell biologists investigating post-transcriptional regulatory mechanisms.

## Introduction

Maintenance of tissue homeostasis requires reciprocal interactions between the extracellular matrix (ECM) and cells that organize their microenvironment. Endothelial cells play a central role in tissue homeostasis and they actively contribute to growth during development, tissue patterning, and regeneration processes in which the remodeling of blood vessel networks is highly dynamic (*Ramasamy et al., 2015*). Fine tuning of interactions between endothelial cells and their perivascular ECM is essential for vascular network formation and integrity (*Marchand et al., 2019*). These interactions occur at integrin adhesion site complexes or 'adhesomes', specialized mechanosensitive hubs for integration of extracellular stimuli and activation of cytoplasmic signaling pathways to control cell adaptive responses (*Humphries et al., 2019*). Large-scale proteomic analyses of integrin adhesomes have revealed the robust presence of RNA Binding Proteins (RBPs) in these macromolecular assemblies

**\*For correspondence:**
delphine.ciais@univ-cotedazur.fr (DC);
ellen.van-obberghen@univ-cotedazur.fr (EVO-S)

**Competing interest:** The authors declare that no competing interests exist.

(*Byron et al., 2015*; *Horton et al., 2016*; *Horton et al., 2015*; *Mardakheh et al., 2015*) yet their precise functions remain to be fully understood.

SAM68 (Src associated in mitosis) is an RBP present in endothelial cell adhesomes (*Atkinson et al., 2018*) whose activities have been associated to cell adhesion and migration (*Huot et al., 2009a*; *Locatelli and Lange, 2011*; *Naro et al., 2022*; *Wu et al., 2015*). SAM68 belongs to the STAR (signal transduction and activation of RNA) family of proteins that link intracellular signaling pathways to RNA processing. First described as a direct target of tyrosine phosphorylation by Src kinase during mitosis (*Fumagalli et al., 1994*; *Taylor and Shalloway, 1994*), SAM68 was subsequently shown to act as a scaffold protein following activation of diverse transmembrane receptors and intracellular signaling pathways (*Sánchez-Jiménez and Sánchez-Margalet, 2013*). A direct role in signal relay has been demonstrated for SAM68 in TNFα signaling following TNFR1 activation (*Ramakrishnan and Baltimore, 2011*), in modulation of the Wnt/β-catenin pathway (*Benoit et al., 2017*) and in Src signaling (*Huot et al., 2009a*; *Richard et al., 2008*).

Moreover, SAM68 has been widely described as a regulator of alternative splicing. As all STAR family proteins, it contains a structural domain for binding of RNA composed of a KH RNA-binding module embedded within a conserved regulatory and signaling region (STAR domain). SAM68 has been shown to regulate the alternative splicing of CD44 in a signal-dependent manner (*Matter et al., 2002*) and the generation of large variants of tenascin-C (*Moritz et al., 2008*). More recently, SAM68 has been identified as regulator of a splicing program involved in migration of triple-negative breast cancer cells (*Naro et al., 2022*). In addition to its major role in alternative splicing, SAM68 impacts other RNA processing events such as transcription, RNA translation, and regulation of long noncoding RNA (*Frisone et al., 2015*).

In light of the signal relay functions of SAM68 and its presence in integrin adhesion complexes, we set out to elucidate the role of SAM68 in endothelial cell-ECM interactions. We surveyed the spatio-temporal distribution of the RBP following integrin activation and analyzed its participation in integrin signaling and adhesion maturation. We show transient localization of SAM68 near nascent adhesion sites where it participates in cytoskeletal remodeling and local delivery of β-actin mRNA, which is known to contribute to adhesion site stabilization and growth. Moreover, we demonstrate that SAM68 orchestrates cell-matrix interactions by regulating the expression of key perivascular ECM components and assembly of the subendothelial matrix, a structure that provides important instructive signals for cell migration and morphogenesis. We propose that these coalescent activities of SAM68 play a key role in the adaptation of endothelial cells to their extracellular environment.

## Results

### SAM68 regulates endothelial cell adhesion site formation and maturation

To determine whether SAM68 is involved in the regulation of the adhesive phenotype of endothelial cells, we performed loss-of-function studies in primary cultured HUVECs by RNA interference. Transfection of two different siRNAs directed against human SAM68 transcripts efficiently decreased SAM68 protein levels (up to 90%), compared to those obsered in control siRNA-transfected cells (*Figure 1—figure supplement 1*). Following an overnight incubation on uncoated glass coverslips, subconfluent cells transfected with the control siRNA displayed large lamellipodial protrusions (*Figure 1A*) and prominent stress fibers (*Figure 1B*). In contrast, cells expressing SAM68-targeting siRNA were less spread and they assembled smaller edge ruffles. Quantitative morphological analyses revealed a significant decrease in the spreading index of SAM68-depleted cells, whereas the relative elongation of these cells was increased (*Figure 1A*). In addition to limited lamellipodial expansion in SAM68-depleted cells, stress fiber formation was affected as can be seen by a marked decrease in the number of stress fibers and a more isotropic arrangement of actin filaments, when viewed at high magnification (*Figure 1B*).

As cytoskeletal organization and cell spreading is intimately linked to the assembly and maturation of cell-substrate attachment sites, we next examined the number and size of integrin adhesions by immunostaining of vinculin, a core component of adhesion complexes, in siRNA-transfected cells. Following overnight plating, significant differences were observed in the abundance, size and distribution of vinculin-positive structures between control siRNA-transfected cells and SAM68-targeting

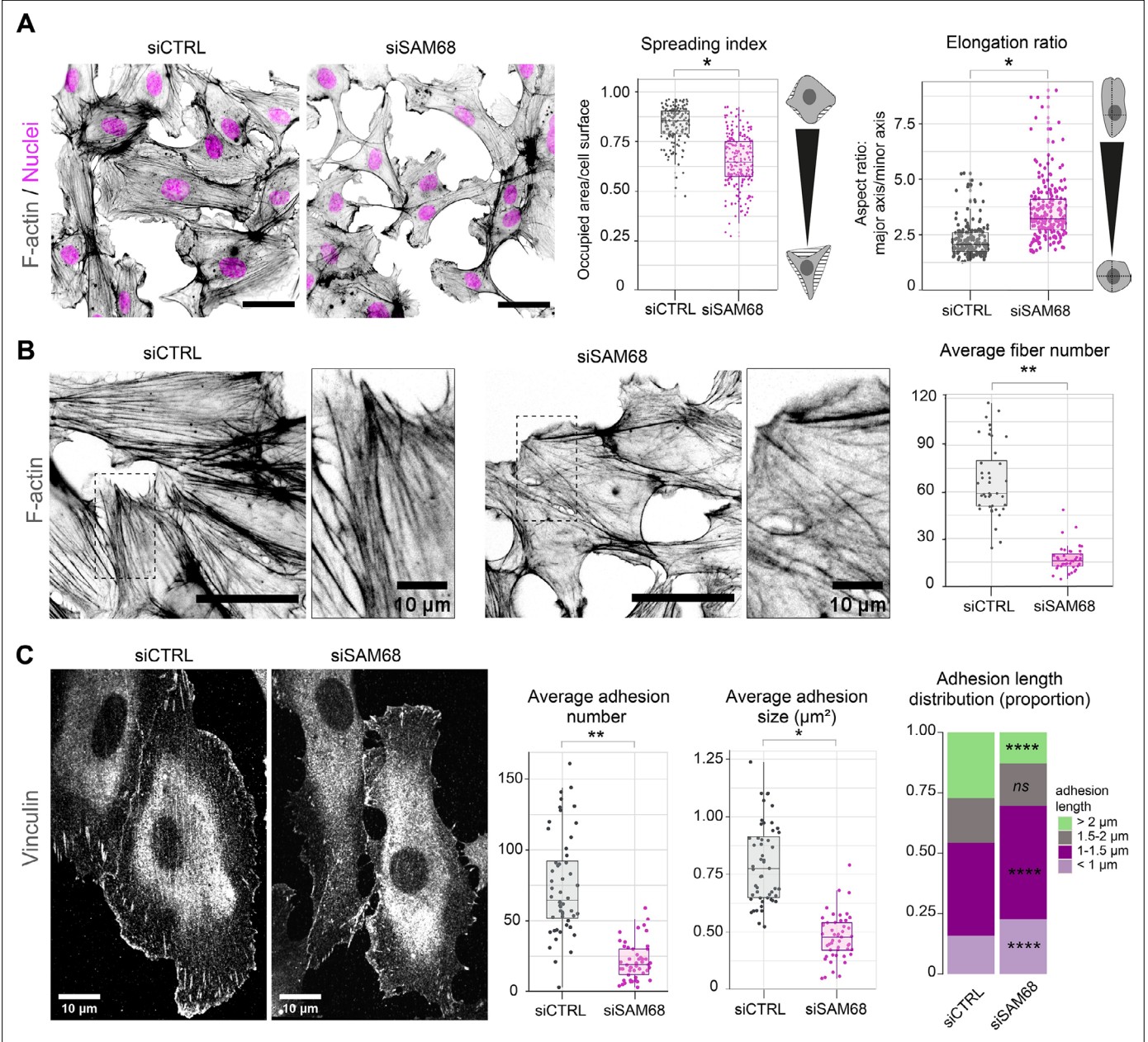

**Figure 1.** SAM68 regulates endothelial cell spreading, formation and maturation of adhesion sites. (**A**) siRNA-transfected cells plated overnight on uncoated glass coverslips were stained for F-actin for cell shape analysis (n=50 cells per condition; N=3). Scale bar=50 µm. Spreading Index is expressed as the ratio of occupied area (hatched area) to the cell surface (solid grey area) and elongation ratio is expressed as the ratio of the major to minor cell axis. (**B**) Magnification images of endothelial cells from the experimental setting described in (**A**). Sacle bars=20 µm and 10 µm (zoomed). Average fiber number was quantified using FIJI software (n=16 cells per condition; N=3). (**C**) Vinculin staining was performed on siRNA-transfected cells plated overnight on glass coverslips. Analysis of adhesion sites was performed using FIJI software by quantifying at least 15 cells per condition (N=3). Statistics: p-values: *<0.05 **<0.01; ****<0.0001. Student's t-test (paired CTRL-siSAM68) was used for (**A, B, C**). Pearson's chi-squared test was used for adhesion length distribution (**C**).

The online version of this article includes the following source data and figure supplement(s) for figure 1:

**Source data 1.** Quantification of cell morphology parameters.

**Source data 2.** Quantification of stress fiber numbers and lengths.

**Source data 3.** Quantification of vinculin-positive structure numbers, lengths and sizes.

**Figure supplement 1.** siRNA based SAM68 depletion in endothelial cells.

**Figure supplement 1—source data 1.** Western blot uncropped membranes.

**Figure supplement 2.** SAM68 depletion decreases adhesion site length.

siRNA-transfected cells (*Figure 1C*). Following SAM68 depletion, the average number of vinculin adhesions was reduced by threefold. Moreover, the average size of vinculin-positive adhesions was decreased by approximatively twofold and adhesions were less elongated (*Figure 1C* and *Figure 1— figure supplement 2*). Adhesion sites are dynamic subcellular structures that have been classified into three main types, depending on their composition, size and distribution (reviewed in *Geiger et al., 2001*). Nascent dot-like *focal complexes* form at the cell periphery and rapidly disassemble or mature into more elongated *focal adhesions,* as force is applied upon linkage of adhesion components to the actin cytoskeleton. Long *fibrillar adhesions* that arise by translocation of α5β1 integrins out of focal adhesions along growing FN fibers on the cell surface are sites of FN fibrillogenesis in mesenchymal cells. Analyisis of the size distribution of vinculin-containing adhesions revealed a decrease in the proportion of long adhesions (>2 µm) in SAM68-depleted cells compared to control cells, and an increase in the proportion of the smallest adhesions (<1 µm; *Figure 1C*). Whereas growth of integrin adhesions to greater than 1 µm is a hallmark of adhesion maturation (*Doyle et al., 2022*), nearly all adhesion sites in SAM68-depleted cells remained shorter than 0.75 µm. The finding that SAM68-depleted cells display fewer, smaller and less elongated adhesion sites, together with the reduced number of actin stress fibers assembled in these cells, suggested that their spreading defect stems from a defect in the formation or maturation of integrin adhesions.

## SAM68 transiently localizes to leading edges of spreading cells

To explore how SAM68 contributes to the stability and growth of endothelial cell adhesion sites, we first examined the localization of the protein in untransfected endothelial cells. SAM68 has previously been shown to relocate near the plasma membrane following fibroblast attachment (*Huot et al., 2009a*). More recently, it was found by mass spectrometry to associate with β3 integrin-based adhesion complexes in endothelial cells plated on FN for 90 min (*Atkinson et al., 2018*). Therefore, we determined SAM68 localization in endothelial cells at early times after plating on FN-coated coverslips. Total internal reflection fluorescence (TIRF) microscopy was used to selectively detect SAM68 near the membrane-coverslip interface (internal depth of ~150 nm) and to minimize the fluorescence signal from nuclear SAM68.

Twenty minutes after seeding, SAM68 was present in dot-like structures at the basal surface of cells. At this early time of adhesion, the majority of the submembraneous SAM68 puncta were concentrated at the edge of spreading cells in, or adjacent to, sites of active cortical actin assembly (*Figure 2A*). Staining of peripheral SAM68 partially overlapped with that of cortactin, a well-known regulator of cortical actin polymerization and substrate of Src kinase (similar to SAM68). Anti-phosphotyrosine staining was also strongest at the periphery of spreading cells, in close proximity to SAM68 puncta (*Figure 2A*).

Live-cell imaging of eGFP-SAM68-expressing HUVEC spreading on a FN-coated substrate between 20 and 45 min revealed that SAM68-containing particles in the peripheral submembraneous compartment were extremely motile (*Figure 2—video 1*). This time-dependent relocalization of SAM68 during cell spreading and focal adhesion formation is illustrated in *Figure 2B*. After 20 min of adhesion, SAM68 dots formed a peripheral ring that partially overlapped with FAK-containing nascent adhesions at cell edges. By 45 min, SAM68-positive puncta became more diffuse as FAK-labeled adhesions became larger and more elongated. After 120 min of cell spreading and adhesion maturation, SAM68 puncta were distributed across the entire basal cell surface and no longer co-localized with FAK-positive focal adhesions. These results regarding the transient localization of SAM68 at the cell periphery, together with our observed effects of SAM68 depletion on cell spreading and adhesion formation, strongly suggest that SAM68 regulates an initial step of adhesion stabilization and maturation.

## SAM68 locally regulates adhesion site signaling

In light of the dynamic and transient association of SAM68 with nascent adhesions at the cell periphery during spreading on ligand-coated coverslips, this experimental setting was not ideal for investigating early recruitment and local functions of the molecule. Therefore, we employed an alternative system for this purpose in which FN-coated beads are deposited onto fully spread endothelial cells following an overnight incubation to generate ectopic integrin adhesion sites at the apical surface of cells, as described (*Atkinson et al., 2018*; *Chicurel et al., 1998*) and schematized in *Figure 3A*. Twenty

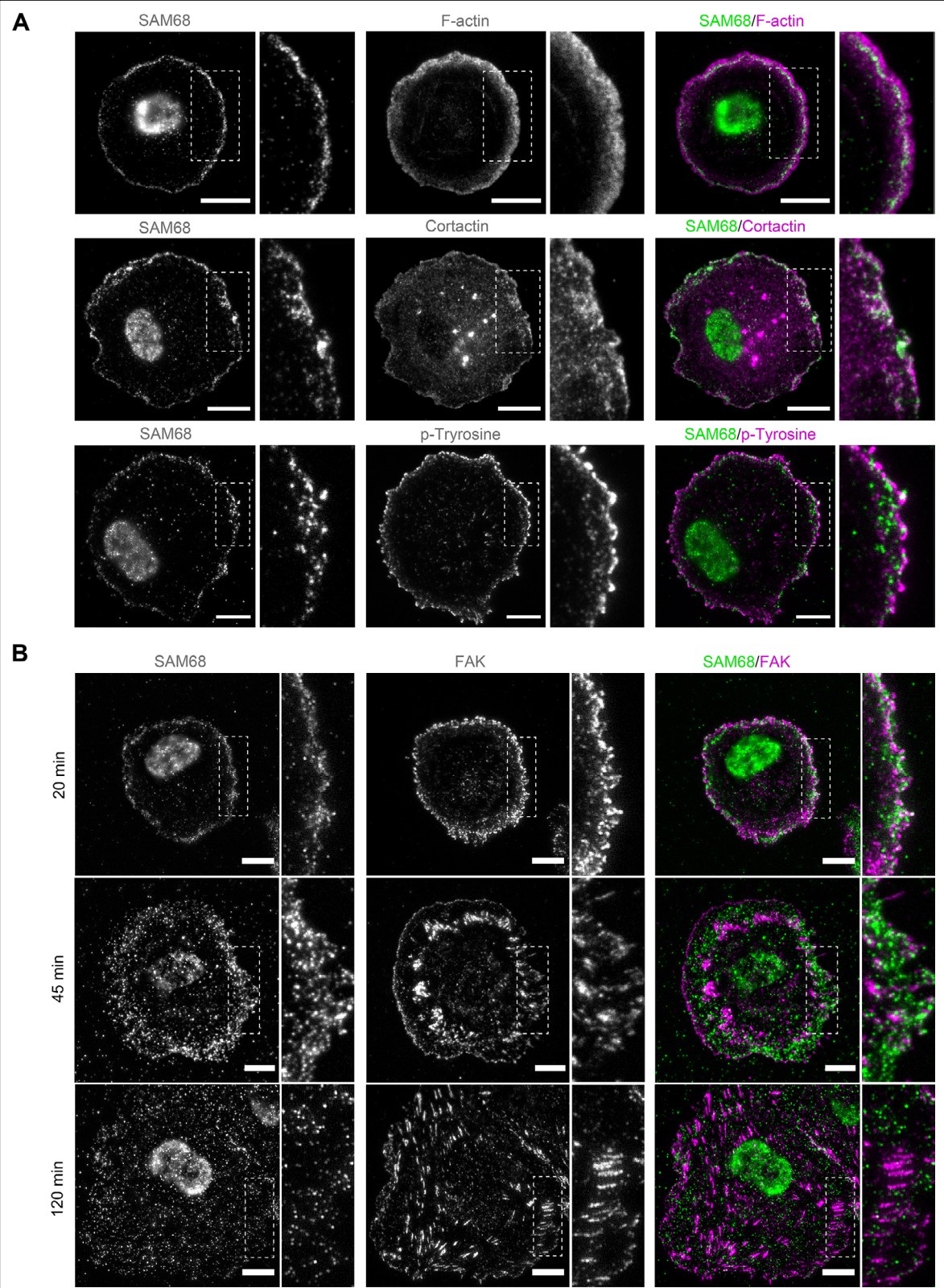

**Figure 2.** SAM68 localization in spreading cells. (**A**) HUVECs plated on FN-coated coverslips for 20 min were stained for SAM68, F-actin, cortactin and phospho-tyrosine. Scale bars=10 μm. Dotted squares depict enlarged areas (10 μm wide) shown in the same panel. (**B**) Labelling of SAM68 and FAK was performed on HUVECs plated on FN-coated coverslips for the indicated times; dotted squares depict enlarged areas shown in the same panel.

*Figure 2 continued on next page*

*Figure 2 continued*

The online version of this article includes the following video for figure 2:

**Figure 2—video 1.** SAM68 transiently localizes at the actin-polymerizing cell front during spreading.

https://elifesciences.org/articles/85165/figures#fig2video1

minutes after addition of beads, the formation of an actin-rich cup containing vinculin was observed at sites of contact with FN-coated beads (*Figure 3B*). Co-staining of vinculin and SAM68 at the apical surface of cells, as shown in the optical sections of *Figure 3B*, illustrates that SAM68 is recruited to these ectopic nascent adhesion sites where it may contribute to some early steps of adhesion complex formation and actin remodeling.

To investigate the function of SAM68 at adhesion sites, we next examined the impact of SAM68 depletion on ectopic adhesion site formation and on integrin-dependent signal transduction initiated at these structures. We first controlled that SAM68-targeted siRNA efficiently diminished SAM68 recruitment to FN-coated beads (*Figure 3—figure supplement 1*). Next, as a readout of integrin signalling, we performed immunolabeling of FAK autophosphorylated on the Src-family kinase binding site tyrosine 397 (pFAK-Y397). pFAK-Y397 is known to be present in both nascent and growing focal adhesions following integrin activation and clustering. As shown in *Figure 3C*, depletion of SAM68 in endothelial cells reduced FAK signaling at ectopic adhesions, as determined by the decreased number of pFAK-Y397-positive foci at the interface of cells with FN-coated beads.

Thereafter, we set out to determine whether the observed effects of SAM68 at adhesion sites could be attributed to its scaffolding activity via interaction with SH3 domain proteins, notably Src (*Taylor and Shalloway, 1994*), or to RNA-processing activities of the protein. To this end, we generated lentiviral constructs encoding wild type SAM68 (SAM68 WT) or the functional mutants depicted in *Figure 3D*. A proline to alanine substitution of residue 358 in the 'P5' SH3 binding domain of SAM68 (SAM68 P358A) has previously been shown to impair Src binding to this domain (*Asbach et al., 2012*), whereas deletion of residues 157–256 (SAM68 ΔKH) results in a full deletion of the SAM68 RNA binding domain and impairment of its binding to RNAs (*Lin et al., 1997*).

Endothelial cells were transfected with an siRNA directed against the 3'UTR of the endogenous SAM68 transcript and then transduced to express either WT or mutant SAM68 coding sequences in a rescue experiment. As shown in *Figure 3D*, expression of SAM68 P358A in endothelial cells depleted for endogenous SAM68 increased the number of pFAK-Y397 foci around beads. In contrast, expression of the RNA binding-defective mutant SAM68 ΔKH drastically decreased integrin signaling, as illustrated by reduced pFAK-Y397 staining at ectopic adhesion sites, and phenocopied the adhesion formation defects observed upon SAM68 depletion. Altogether, these results indicate that both scaffolding and RNA binding activities of SAM68 are required to modulate integrin signaling.

## SAM68 locally delivers mRNA to nascent adhesion sites

The maturation of integrin adhesions is a finely regulated process which involves not only recruitment and scaffolding of adhesome complex components, but also linkage of the growing adhesions to polymerzing actin networks for force transmission. Interestingly, it has been shown that β-actin mRNA localization at focal adhesions contributes to adhesion stability (*Katz et al., 2012*). Importantly, β-actin mRNA is a bona fide RNA target of SAM68 (*Itoh et al., 2002*; *Klein et al., 2013*) and SAM68 has recently been identified as belonging to the RBP proteome recruited onto β-actin mRNA (*Mukherjee et al., 2019*). Moreover, the dot-like pattern of SAM68 staining at adhesions is reminiscent of ribonucleoprotein (RNP) particles. Thus, using the ectopic adhesion assay we futher explored potential RNA binding functions of SAM68 at endothelial cell adhesion sites. To do so, we first tested whether β-actin mRNA is recruited to ectopic integrin adhesions upon cell binding to FN-coated beads. As depicted in *Figure 4A*, RNA smFISH revealed punctuate β-actin mRNA staining around beads. Interestingly, β-actin mRNA particles partially overlapped with those containing SAM68, suggesting that they could be co-localized in the same RNA-protein assemblies. Next we evaluated the implication of SAM68 in the local delivery of β-actin mRNA to nascent apical adhesions. Indeed, upon SAM68 depletion the number of β-actin mRNA foci around beads specifically decreased, as compared to control cells, with no change of foci density in the cytoplasm (*Figure 4B*).

To test whether the β-actin mRNA localizing activity of SAM68 is direct, we used antisense blocking oligonucleotides (chimeric 2'-O-methyl DNA oligos) antisense to the Sam68-binding sequence of

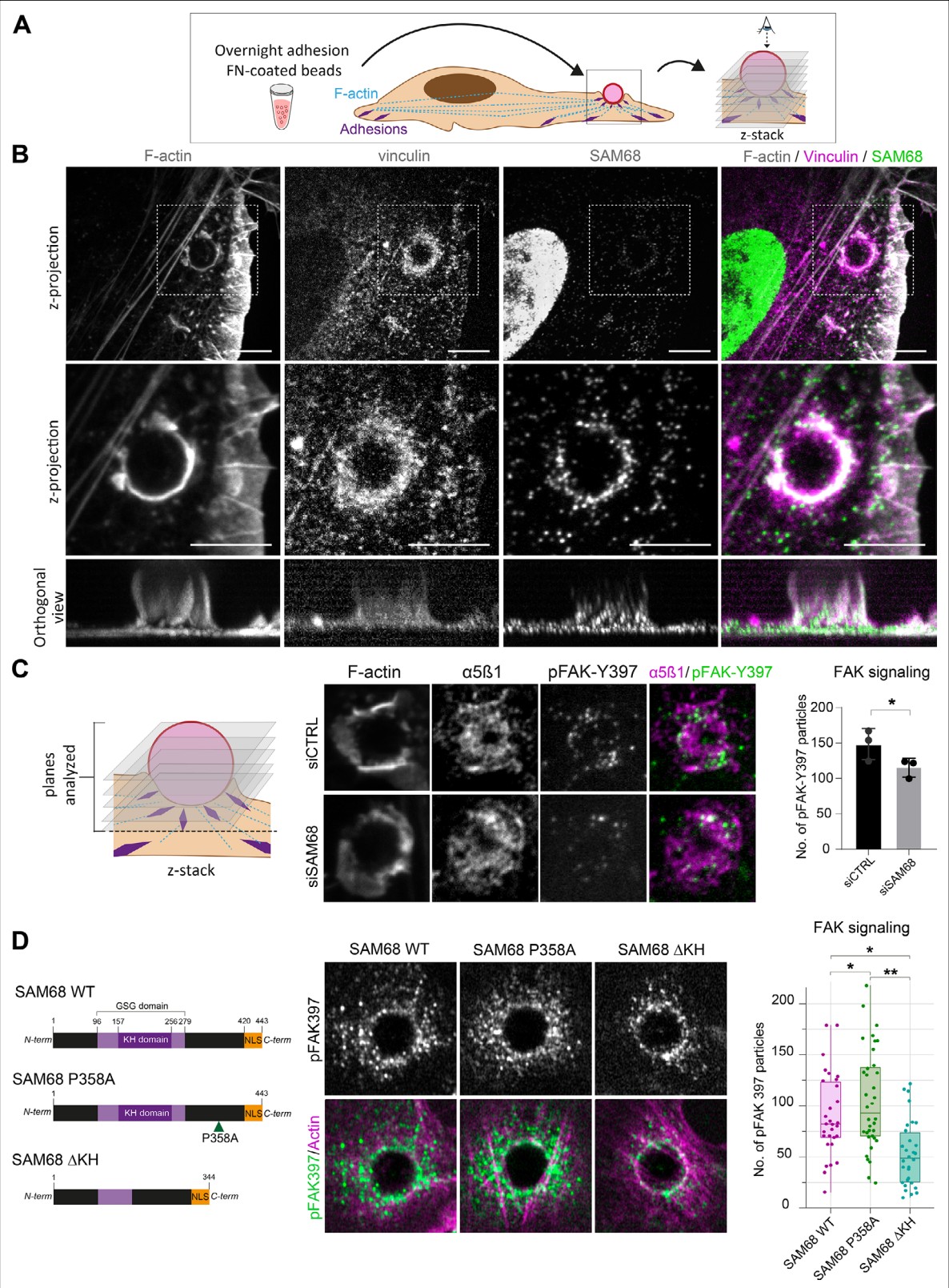

**Figure 3.** SAM68 regulates integrin signaling and RNA composition at adhesion sites. (**A**) Scheme of the experimental procedure used to induce and image artificial adhesion sites in contact with FN-coated beads. (**B**) Labelling of SAM68 and vinculin was performed 20 min after seeding cells on FN-coated beads. Dotted squares in top panels depict enlarged z-projections shown in the middle panel. Orthogonal views are shown in bottom panels. Scale bars=5 µm. (**C**) Immunolabeling of α5β1 and activated FAK (pFAK-Y397) were performed 20 min after deposition of FN-coated beads

*Figure 3 continued on next page*

*Figure 3 continued*

onto siCTRL- or siSAM68-transfected cells and pFAK-Y397 foci were quantified (n=at least 8 beads per condition, N=3). (**D**) siSAM68-transfected cells were transduced with lentiviral constructions encoding SAM68 WT and mutants shown in the left panel of the figure. Immunolabeling of activated FAK (pFAK-Y397) was performed 20 min after deposition of FN-coated beads onto cells and pFAK-Y397 foci were quantified (n=at least 5 beads per condition, N=4). Statistics: p-values: *<0.05 **<0.01. Student's t-test (paired CTRL-siSAM68) was used for (**C, D**).

The online version of this article includes the following source data and figure supplement(s) for figure 3:

**Source data 1.** Quantification of pFAK397 foci from *Figure 3C*.

**Source data 2.** Quantification of pFAK397 foci from *Figure 3D*.

**Figure supplement 1.** SAM68-targeted siRNA efficiently diminishes SAM68 recruitment to FN-coated beads.

β-actin mRNA (*Itoh et al., 2002*), previously shown to disrupt binding between SAM68 its β-actin mRNA cargo in dendrites (*Klein et al., 2013*). In endothelial cells, transfection of the blocking oligo-nucleotides #SBE1 and #SBE2, indicated in *Figure 4C*, resulted in a decrease in the recruitment of β-actin mRNA particles around beads, as compared to their recruitment in control (scrambled) oligonucleotide-transfected cells. Importantly, there was no change of foci density in the cytoplasm. These results confirm the involvement of SAM68 in β-actin mRNA delivery to adhesion sites and indicate that β-actin transcript recruitment is likely caused by direct binding of SAM68 to the 3′ UTR of β-actin mRNA.

## SAM68 modulates FN synthesis and fibrillogenesis

In endothelial cells, stabilized focal adhesions elongate and mature into fibrillar adhesions spanning the ventral cell surface. The accumulation of fibrillar adhesions can be visualized by immunostaining of integrin α5β1 (*Figure 5A*). Consistent with the observed role of SAM68 in adhesion maturation, we detected a significant decrease in the number and length of fibrillar adhesions in SAM68-depleted cells, as shown in *Figure 5A*.

Fibrillar adhesions are sites of FN fibrillogenesis. Thus, the sparsity of fibrillar adhesions in SAM68-depleted cells prompted us to investigate the ability of these cells to deposit a FN matrix. As illustrated by immunostaining of FN in endothelial cells plated on uncoated glass coverslips (*Figure 5B*), depletion of SAM68 markedly perturbed FN deposition. SAM68 knock down affected not only the amount of FN deposited beneath cells, but also perturbed the organization of the assembled protein. Thus, FN associated to SAM68-deficient cell monolayers was mostly present in the form of aggregates or thick cables, as opposed to the more homogeneous thin fibrillar networks assembled by control cells, as seen in intensity profiles of FN labeling along the lines indicated in *Figure 5C*. We did not observe differential retention of FN in the cytoplasm of SAM68-depleted cells. Rather, FN staining was strictly fibrillar (ECM-associated) in both control and SAM68-depleted cells and the intensity profile baseline values were similarly low, indicating that misregulation of FN deposition does not result from altered secretion of the molecule.

FN assembly by endothelial cells was previously shown to be tightly coupled to autocrine production of the protein (*Cseh et al., 2010*). Therefore, we examined the possible role of SAM68 in the regulation of FN expression. Indeed, western blot analysis of total cell lysates revealed an approximately 50% decrease in the levels of cell-/matrix-associated FN in SAM68-depleted cells, as compared to control cells (*Figure 5D*). SAM68 depletion also led to a decrease in soluble FN in cell conditioned medium (*Figure 5—figure supplement 1*). As these results clearly indicated that SAM68 modulates production of the protein, we next evaluated the effect of SAM68 depletion on FN transcript levels (*Figure 5E*). Cellular FN differs from plasma FN by the inclusion of one or both 'Extra Domains' EDB and EDA (namely oncofetal FN isoforms) by alternative splicing. Therefore, we designed PCR primer pairs to specifically detect total FN transcripts (tFN) or FN transcripts containing sequences encoding the EDB- and/or EDA domains, which have been shown to differentially affect FN fibrillogenesis and endothelial cell behavior (*Cseh et al., 2010*; *Efthymiou et al., 2021*). Upon depletion of SAM68 in endothelial cells, a robust downregulation of total and Extra Domain-containing isoforms of FN was observed. Global downregulation of all FN transcripts in siSAM8-depleted cells suggested that SAM68 might affect *FN1* gene transcription. To test this, we generated a reporter construct (pFN) containing a 3.5 kb sequence encompassing the Src signaling-responsive *FN1* gene promoter region (*Dean et al., 1987*) upstream of the firefly

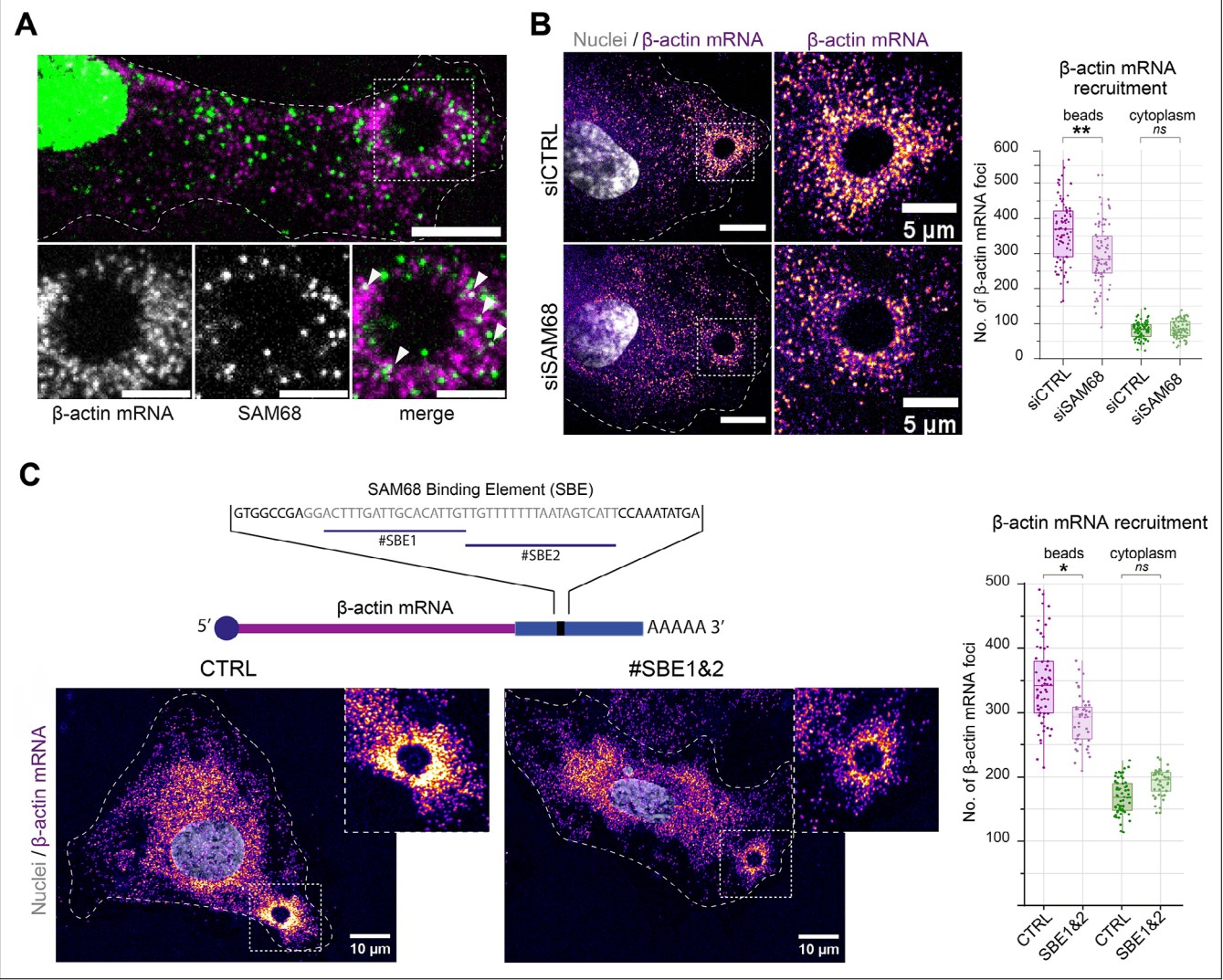

**Figure 4.** SAM68 is involved local delivery of β-actin mRNA at adhesion sites. (**A**) smRNA FISH and SAM68 stainings were performed 20 min after deposition of FN-coated beads. Scale bars top image=10 μm, enlarged area 5 μm. Arrowheads point to overlapping signals between β-actin mRNA and SAM68 protein. (**B**) smRNA FISH of β-actin performed on cells 20 min after addition of FN-coated beads to cultures of siCTRL- or siSAM68-transfected cells (n=at least 12 beads per condition, N=3). Scale bars = left 10 μm, enlarged area 5 μm. (**C**) smiRNA FISH staining of β-actin performed on cells 20 min after deposition of FN-coated beads onto endothelial cells transfected with CTRL or blocking oligonucleotides (#SBE1 and #SBE2, as indicated) (n=at least 12 beads per condition, N=3). Statistics: p-values: *<0.05 **<0.01. Student's t-test (paired CTRL-siSAM68) was used.

The online version of this article includes the following source data for figure 4:

**Source data 1.** Quantification of pFAK397 foci from *Figure 4B*.

**Source data 2.** Quantification of pFAK397 foci from *Figure 4C*.

luciferase coding sequence. SAM68-regulated promoter activity was examined by co-transfecting the pFN construct with increasing concentrations of a plasmid encoding SAM68 (pSAM68), or an empty vector (pcDNA3) control. Experiments were performed in HEK293 cells, since endothelial cells are poorly transfectable. In addition to their high transfection efficiency, HEK293 cells display nearly undetectable expression of FN and they are unable to assemble the molecule (even upon FN overexpression see *Efthymiou et al., 2021*). Luciferase measurements showed that increasing the amount of pSAM68 transfected in these cells augmented *FN1*-driven luciferase activity, compared to that induced by pcDNA3 transfection (*Figure 5F*). Notably, transfection of as little as 25 ng of pSAM68 yielded a 1.5-fold increase in luciferase activity, compared to cells co-transfected with the control plasmid, attesting to the involvement of SAM68 in *FN1* promotor regulation. We confirmed the involvement of SAM68 in *FN1* gene transcription in endothelial cells and characterized the

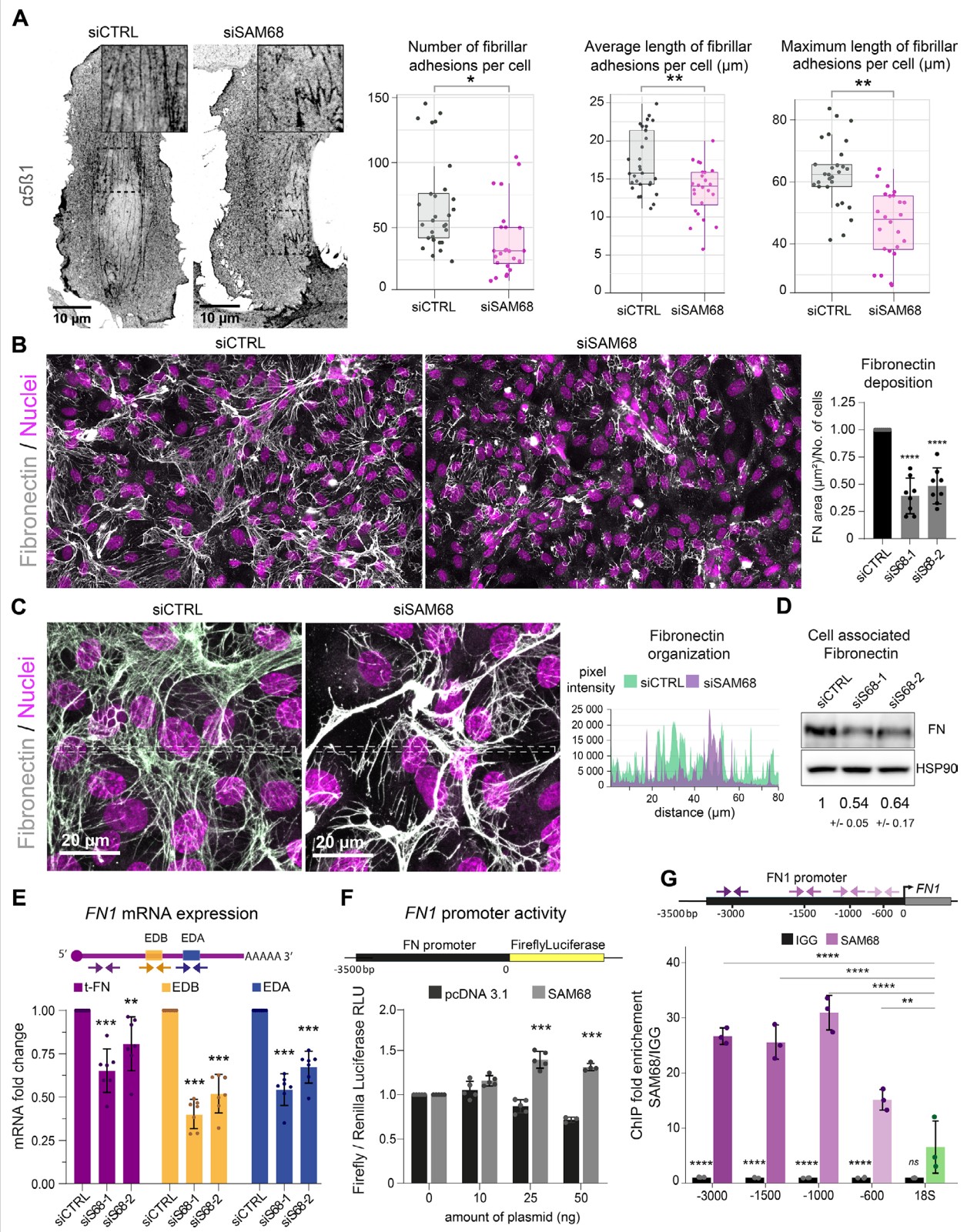

**Figure 5.** SAM68 is involved in FN assembly and expression. (**A**) Immunofluorescence staining of α5β1 integrin was performed to identify and quantify fibrillar adhesion in siRNA transfected cells plated overnight on glass coverslips (n=at least 15; N=3). (**B**) Immunofluorescence staining of FN was performed on siRNA transfected cells plated on glass coverslips and quantification on whole-coverslip scans is expressed as the ratio of FN-stained area to the number of cells (N=8). Representative 40x field views. (**C**) Representative high magnification images of FN staining are shown with areas between

*Figure 5 continued on next page*

*Figure 5 continued*

the dotted lines selected for fluorescence intensity profiles. (**D**) Western blot analysis of cell-associated FN in siRNA transfected cells with densitometric quantification indicated below (N=3). (**E**) qPCR analysis of total FN (tFN) and Extra Domain-containing isoform expression in siRNA transfected cells using the indicated qPCR primer pairs (N=7). (**F**) Measurements of Luciferase activity driven by the *FN1* promoter when SAM68 is overexpressed (N=5). (**G**) DNA fragments located in the *FN1* promoter were quantified by qPCR in anti-SAM68 or IgG immunoprecipitated complexes (N=3). Statistics: p-values: *<0.05 **<0.01 ***<0.001 ****<0.0001. Student's t-test (paired CTRL-siSAM68) was used for (**A, B, D, E, F**). Statistical analysis of fold enrichment in (**G**) was performed with R using pairwise t-test with p-values adjusted using 'Bonferroni correction'.

The online version of this article includes the following source data and figure supplement(s) for figure 5:

**Source data 1.** Quantification of endothelial fibrillar adhesions.

**Source data 2.** Western blot uncropped membranes.

**Source data 3.** Quantification of *FN1* mRNA levels.

**Source data 4.** Quantification of *FN1* promotor reporter activity.

**Source data 5.** Quantification of SAM68 protein recruitment onto the *FN1* promotor.

**Figure supplement 1.** SAM68 depletion decreases insoluble ECM-associated FN and soluble FN secreted in cell culture medium.

**Figure supplement 1—source data 1.** Western blot uncropped membranes.

**Figure supplement 2.** SAM68 contributes to the alternative splicing of FN transcripts.

recruitment of SAM68 to the endogenous *FN1* promoter by performing ChIP experiments. qPCR quantifications shown in *Figure 5G* reveal that SAM68 specifically associates with different regions of the *FN1* promoter in these cells, but not with genomic ribosomal DNA. In aggregate, these results indicate that the presence of SAM68 positively regulates FN expression by increasing transcription of the *FN1* gene.

As SAM68 is a member of the STAR family of mRNA processing proteins and is known to regulate the alternative splicing of mRNAs encoding several cellular proteins, we analyzed the contribution of SAM68 to alternative splicing of FN transcripts. To do so, the relative expression of specific oncofetal FN transcripts (FN EDA +or FN EDB+) was normalized to the expression of total FN (tFN) transcripts (*Figure 5—figure supplement 2*). Selective downregulation of each Extra Domain-containing isoform was observed following SAM68 depletion, suggesting that SAM68 is also involved in the regulation of FN transcript diversity via alternative splicing during gene expression.

## SAM68 depletion alters endothelial cell basement membrane composition and biogenesis as well as cell-ECM interactions

Considering the major impact of SAM68 depletion on FN production and fibrillogenesis, and the well-documented role of FN on the assembly of higher order ECM networks, we next addressed the role of SAM68 in regulating the expression and organization of other components of the endothelial cell basement membrane. As shown in *Figure 6A*, the mRNA levels of selected matrisome components known to be involved in subendothelial matrix deposition and/or maturation (*Marchand et al., 2019*) were quantified. Interestingly, SAM68 depletion had no impact on COL4A1, LAMA4, NID1, or HSPG2 (perlecan) transcript levels, but it significantly decreased expression of those encoding the COL8A1 subunit of type VIII collagen, periostin, fibulin-1, and biglycan. Even though SAM68 depletion had no direct effect on the levels of COL4A1 and HSPG2 transcripts, incorporation of the corresponding proteins (assuming that they are similarly translated) into the matrix was severely compromised, as shown in *Figure 6B*, and quantified in *Figure 6C*.

This role of SAM68 in subendothelial matrix synthesis and assembly suggested that SAM68 may participate in matrix-dependent modulation of endothelial cell adhesive behavior. To evaluate this, we compared the adhesion of control or SAM68-depleted cells to their own matrix by subjecting confluent monolayers to repeated rounds of mild trypsin digestion. siRNA-transfected cells were plated for 48 hr prior to protease treatment. Enumeration of detached cells in each fraction of diluted trypsin revealed that 80% of the SAM68-depleted cells detached from the matrix within 10 min of treatment whereas only 40% of control siRNA-transfected cells were detached at this time (*Figure 6D*). Taken as a whole, these results indicate that in addition to exerting local effects on cell adhesion complexes, SAM68 regulates endothelial cell matrisome composition, abundance and cell-matrix interactions.

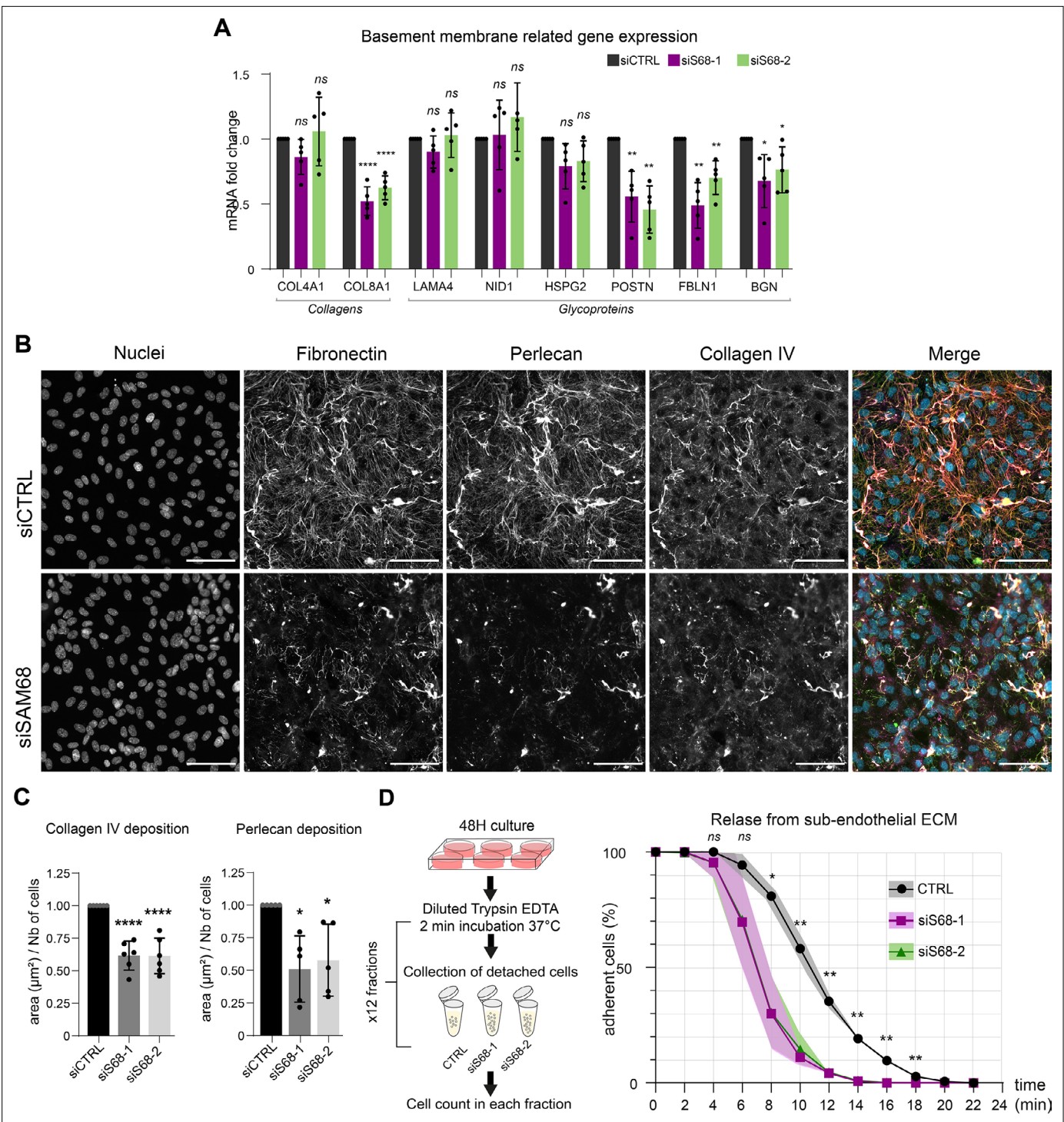

**Figure 6.** SAM68 regulates ECM protein deposition and mRNA biogenesis of matrisome genes. (**A**) qPCR analysis of selected mRNA expression in siRNA transfected cells (N=5). (**B**) Representative images of collagen IV and perlecan staining of siRNA transfected cells 48 hr after plating on uncoated glass coverslips. Scale bars=50 μm. (**C**) Quantification of collagen IV and perlecan staining (N=5). (**D**) (left) Schematic of detachment assay used to determine release of siRNA-transfected cells from their self-assembled ECM support. (right) Quantification of adherent, trypsin-resistant cells (N=3). Statistics: p-values: *<0.05 **<0.01 ***<0.001 ****<0.0001. Student's t-test (paired CTRL-siSAM68) was used for (**A, C, D**).

The online version of this article includes the following source data for figure 6:

**Source data 1.** qPCR quantification of basement membrane components.

**Source data 2.** Quantification of ECM protein staining area.

**Source data 3.** Quantification of attached cells.

## SAM68 depletion enhances cell migration but impairs endothelial cell sprouting

Proper organization and scaffolding of matrix components in the endothelial basement membrane are required for efficient capillary formation during angiogenesis and for maintenance of vascular homeostasis. Thus, we next assessed the functional consequences of SAM68 depletion on the motile behavior of endothelial cells in 2 and 3D contexts. In a 2D setting, migration of individual cells in sparse cultures was followed by time lapse video microscopy. As shown in *Figure 7A* and *Figure 7—figure supplement 1*, both the velocity and the total distance travelled by SAM68-depleted cells was increased, as compared to control cells. Collective cell migration was also affected, as determined in a wound-healing assay in which SAM68 depletion significantly increased the speed of wound closure (*Figure 7B*). The observed increment in motility of siSAM68-transfected cells could be at least partially explained by the decrease in FN expression/assembly that tends to restrict cell movements (*Cseh et al., 2010*; *Serres et al., 2014*).

We next interrogated the regulation of angiogenic behavior of endothelial cells by SAM68 in a 3D sprouting assay. To do so, Cytodex3 beads fully covered with siCTRL- or siSAM68-transfected cells were embedded in fibrin gels for 48 hr prior to confocal imaging of sprout formation. As shown in *Figure 7C*, control cells formed multiple elongated sprouts with tip cells exhibiting abundant filopodial extensions. Quantitative image analysis revealed that SAM68 depletion had no effect on the number of sprouts per bead whereas it significantly decreased the length of each sprout *Figure 7* and *Figure 7—figure supplement 2*. Closer inspection at the tip of invading cords revealed that SAM68-deficient cells displayed fewer filopodia (zoomed insert, *Figure 7C*), reflecting defects at adhesion sites, and detached more readily from neighboring cells than control cells (white arrows), suggesting that compromised FN assembly and ECM production weakened the cohesiveness of these multi-cellular alignments. Altogether, these data suggest that SAM68 contributes to capillary morphogenesis of endothelial cells.

## Discussion

Here, we report that the RNA binding protein SAM68 regulates spreading, migration and the angiogenic phenotype of endothelial cells. First, we showed that SAM68 is localized near polymerizing actin in spreading cells where it contributes to the stabilization and growth of nascent integrin adhesions by local regulation of integrin signaling and delivery of β-actin mRNA transcripts. Further, at the cellular and multi-cellular levels we found that SAM68 contributes to the conditioning of the adhesive environment of cells by enhancing the expression and/or deposition of principal vascular basement membrane components which, in turn, affect cell motility and capillary-like formation. These findings demonstrate an important link between SAM68 and the angiogenic phenotype of cultured endothelial cells via coordinated functions of the RBP at submembraneous adhesion sites and in the nucleus, as schematized in *Figure 8*.

## SAM68 is involved in integrin signal transduction and RNA localization during cell adhesion formation and maturation

SAM68 localization at the plasma membrane was previously documented in mouse embryo fibroblasts (MEFs) during early adhesion on a FN-coated substrate (*Huot et al., 2009a*). However, in this pioneering study SAM68-deficient cells displayed an accelerated spreading phenotype, rather than the impaired spreading phenotype that we observed in endothelial cells following SAM68 knockdown. These apparently divergent results could stem from phenotypic variability of the MEF clones used in Huot et al., as experiments were conducted on MEF isolates from SAM68$^{+/+}$ or SAM68$^{-/-}$ mice, as opposed to using a loss-of-function approach on the same cell population as reported here. In subsequent work, SAM68 was detected at the spreading edge MRC5 fibroblasts in actin-sheathed bleb-like structures termed Spreading Initiation Centers (SIC) (*Bergeman et al., 2016*). These structures containing multiple RBPs, as determined by quantitative mass spectrometry, were proposed to be involved in regulation of early cell adhesion processes and spreading (*de Hoog et al., 2004*). Although we did not observe SIC-like blebs in endothelial cells spreading on FN substrates, SAM68 was transiently recruited to the cell membrane in highly motile punctate structures that localized near sites of cortical actin assembly where nascent integrin-based adhesions form. Indeed, SAM68 is a bona

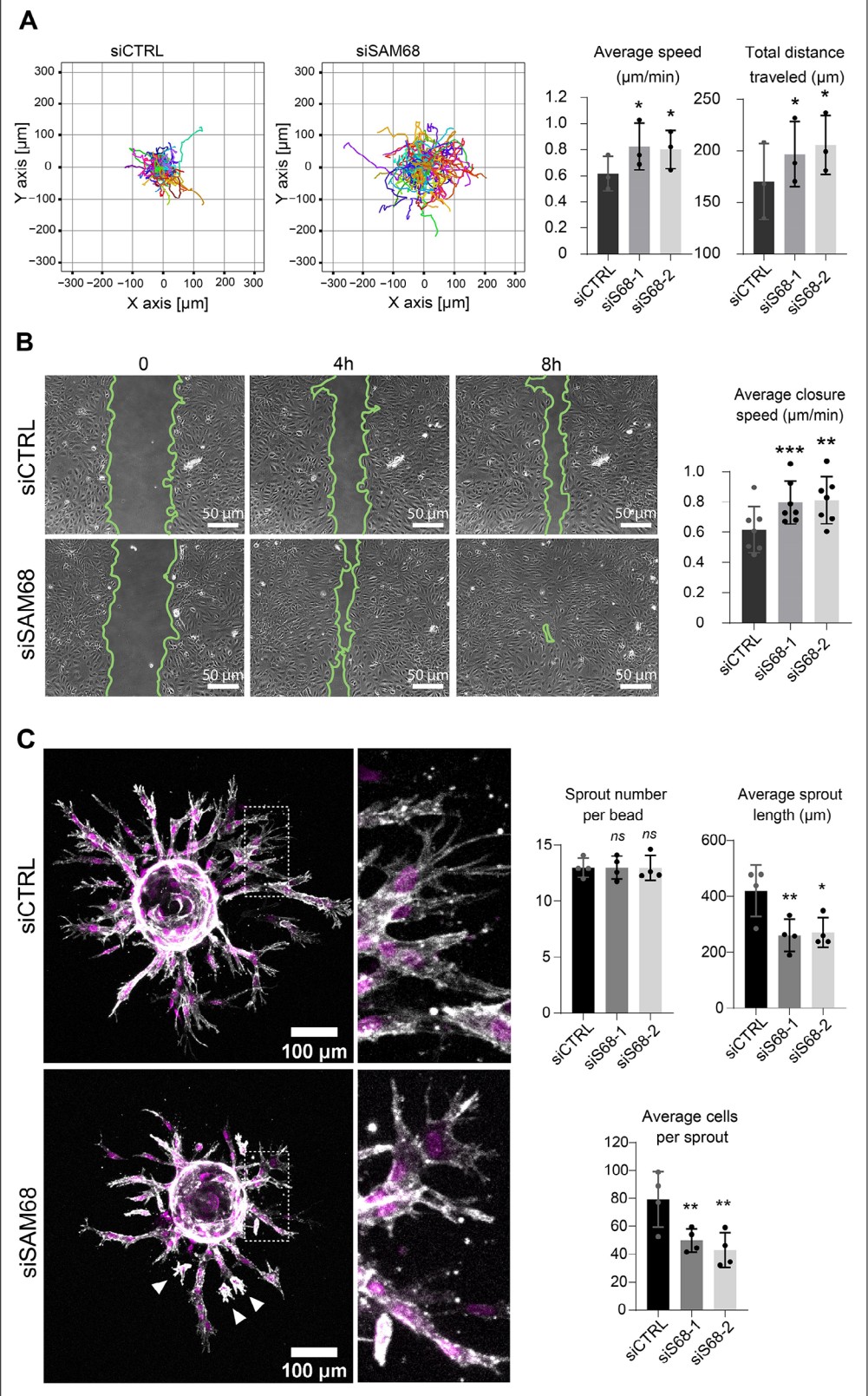

**Figure 7.** SAM68 depletion regulates endothelial cell migration and angiogenic sprouting behavior. (**A**) Migration of individual siRNA-transfected cells was analyzed by time lapse microscopy. Individual tracks of one representative experiment are plotted and cell velocity as well total distance travelled are quantified (N=3). (**B**) Representative images from a wound assay experiment on siRNA transfected cells are shown together with quantification of

*Figure 7 continued on next page*

*Figure 7 continued*

average closure speed (N=7). (**C**) (left) siRNA-transfected cells on Cytodex3 beads embedded in a fibrin gel for 48 hr prior to staining for F-actin (gray) and nuclei (magenta). Cells detached from the sprouting structures are indicated with arrowheads. Quantification of sprout characteristics is shown on the right (n=12 Cytodex3 beads per condition, N=4). Statistics: p-values: *<0.05 **<0.01 ***<0.001. Student's t-test (paired CTRL-siSAM68) was used for (**A, B, C**).

The online version of this article includes the following source data and figure supplement(s) for figure 7:

**Source data 1.** Quantification of individual cell tracks.

**Source data 2.** Quantification of average closure speed.

**Source data 3.** Quantification of endothelial sprouting.

**Figure supplement 1.** SAM68 depletion regulates endothelial cell migration.

**Figure supplement 2.** SAM68 depletion regulates endothelial cell angiogenic sprouting behavior.

fide component of FN-induced integrin adhesion complexes, identified in endothelial cell adhesomes (*Atkinson et al., 2018*) and in meta-analysis of mass spectrometry datasets generated from different laboratories using multiple cell types (*Horton et al., 2015*). Here, using an ectopic adhesion assay we provide definitive proof that SAM68 is recruited to adhesions upon integrin engagement and we demonstrate that it participates in integrin signaling through the modulation of FAK phosphorylation on tyrosine 397, a critical site for c-Src binding and activation of FAK itself. Direct interactions between SAM68 and intrinsic components of the adhesome such as FAK (*Yi et al., 2006*), tensin3 (*Qian et al., 2009*) and Csk (C-terminal Src kinase) have been reported (*Huot et al., 2009a*). In the case of Csk, association of tyrosine-phosphorylated SAM68 with the endogenous inhibitor of c-Src following adhesion to FN was found to induce c-Src downregulation, thereby linking SAM68 to the transient nature of Src activation. This is in line with our results that interfering with SAM68-c-Src interactions by

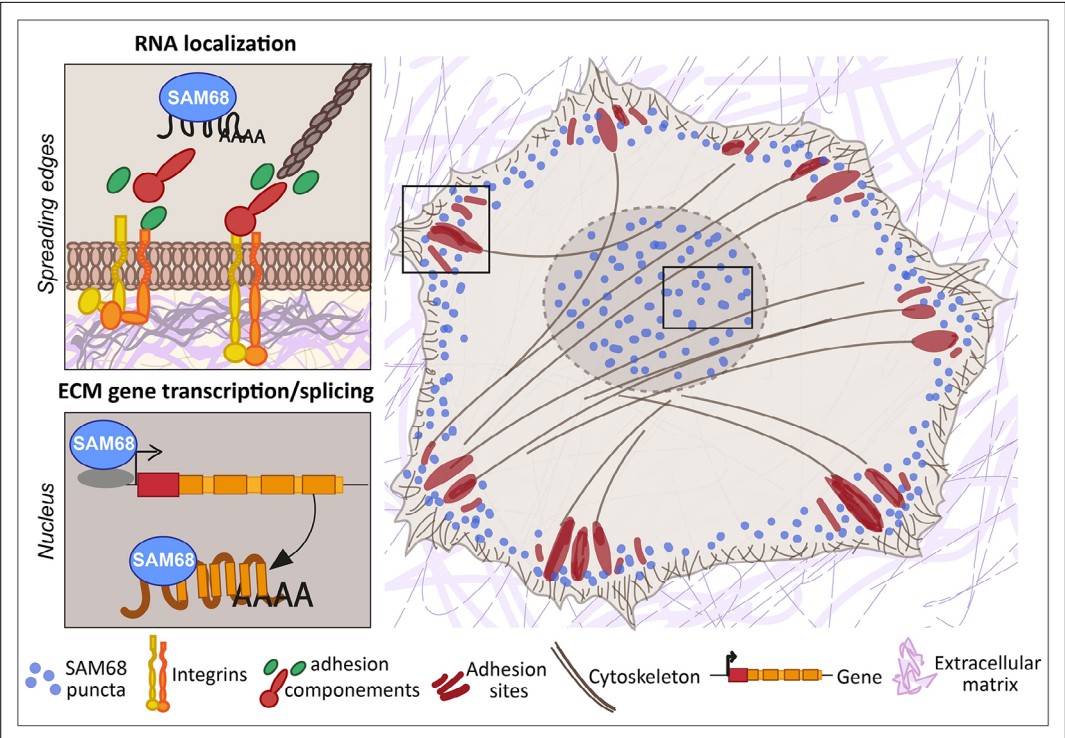

**Figure 8.** SAM68 regulates integrin adhesion maturation and matrix conditioning in endothelial cells. Upon integrin activation, a cytoplasmic fraction of SAM68 participates transiently in adhesion complex stabilization, through regulation of integrin signaling and localization of β-actin mRNA. In the nucleus, SAM68 concomitantly regulates the expression of key subendothelial matrix genes, thereby promoting basement membrane assembly and conditioning. These coalescent functions of SAM68 enhance endothelial cell adaptation to their microenvironment and point to an important role for SAM68 during angiogenesis.

expression of the of SAM68 P358A led to a slight increase in pFAK-Y397 foci number around beads possibly due to sustained Src activity. The activated FAK/c-Src complex phosphorylates downstream signaling proteins to promote actin assembly. One of the principal defects of SAM68-depleted endothelial cells is their inability to convert nascent adhesions to mechanically loaded focal adhesions. This conversion proceeds by recruitment of adhesome components, including actin-associated proteins that regulate actin assembly and generate mechanical tension required for cell shape changes as well as locomotion.

The group of Singer showed that localization of β-actin mRNA near cell attachment sites by the RBP ZBP1 contributes to the stability of focal adhesions (*Katz et al., 2012*), and they hypothesized that local translation of these mRNAs enhances the association between adhesions and newly synthesized actin filaments. Here we extend this finding by showing that the RBP SAM68 also contributes to the delivery of β-actin mRNA to ectopic adhesions. At these sites, SAM68 particles were transiently enriched near nascent adhesions in close proximity to actin filaments, in what has been described as the actin regulatory layer (*Kanchanawong et al., 2010*). This distribution is of particular interest since SAM68 binding to multiple RNAs encoding adhesome linker or actin-binding proteins can be found in CLIP datasets encode project ENCSR628IDK from (*Van Nostrand et al., 2016*). Whereas these interactions and their relevance for endothelial cell functions remain to be confirmed and elucidated, we speculate that selective RNA localization by SAM68 is an underappreciated step in focal complex stabilization and growth. Altogether, these results are in line with the proposed role for SAM68 in localized mRNA translation during cell adhesion (*Bergeman et al., 2016*) and strengthened by the demonstration by (*Willett et al., 2010*) that translational machinery components are recruited to β3 integrin-enriched adhesion sites in leading edge ruffels of spreading cells.

## SAM68 regulates cell locomotion through conditioning of the ECM environment

Cell migration involves a complex interplay between integrin signaling and ECM composition. The involvement of SAM68 in cell migration was previously addressed in studies with varying outcomes, depending upon the cell type examined and experimental design. In epithelial cells and fibroblast-like synoviocytes, SAM68 silencing was reported to decrease migration and invasion (*Huot et al., 2009b*; *Lin et al., 2022*; *Locatelli and Lange, 2011*; *Sun et al., 2018*) whereas in our study SAM68 depletion in endothelial cells was found to enhance cell migration. This apparent discrepancy highlights the fact that SAM68 functions are cell type dependent and highly contextual. Indeed, different cell types display distinct integrin repertoires and actin microfilament organization. The ECM that they produce and cell-type-specific signaling pathways that regulate actin remodeling and ECM gene expression can also be quite diverse. With respect to experimental conditions, most previously reported cell migration assays have been performed on adsorbed matrix proteins such as FN, Matrigel or collagen type I which can bypass the effects of autocrine ECM production. In contrast, our 2D migration assays were performed using uncoated culture dishes onto which the endothelial cells deposited their own 'autocrine' matrix that restrains their movement. Indeed, we have previously shown in endothelial and glioblastoma (mesenchymal-like) cells that 2D migration is inversely related to the quantity of FN produced by cells (*Cseh et al., 2010*; *Serres et al., 2014*). In both cell types, when matrix deposition was compromised by the knockdown of FN, cells moved more rapidly and more persistently. Moreover, the extent of FN fibrillogenesis is positively correlated to FN expression (*Radwanska et al., 2017*). Thus, in the case of SAM68 depletion, the increased cell motility observed in the present study can be attributed, at least in part, to reduced expression and deposition of autocrine FN, which has repercussions on the integration of other matrix components in the ECM.

Regarding the molecular mechanisms underlying FN expression in endothelial cells, we found that SAM68 can regulate transcription of the *FN1* gene by interacting directly with its promoter. Similarly, in a recent study, the RBP FUS was shown to induce expression of collagen IV by binding to its promoter (*Chiusa et al., 2020*). Although SAM68 is not a transcription factor per se, an increasing number of studies have reported that Sam68 can interact with several major transcription factors (e.g. p300, NF-kB), as reviewed in (*Frisone et al., 2015*). Hence, SAM68 could also regulate *FN1* gene transcription through indirect interactions with its promoter.

In addition to its transcriptional activities, SAM68 is a well-known regulator of alternative splicing. However, we were unable to detect a direct effect of the RBP on alternative inclusion of exons encoding

EDB and/or EDA using mini-gene reporters (D Ciais, personal communication, 02/2018), suggesting that SAM68 regulates isoform diversity through its transcriptional activity. Indeed, functional coupling between splicing and transcription of *FN1* has been reported for EDA inclusion whereby promoter structure and strength were shown to regulate exon inclusion (*Cramer et al., 1997*). The notion of isoform diversity is of particular importance for FN, as Extra Domain-containing isoforms display both distinct and overlapping functions (reviewed in *Efthymiou et al., 2020*; *White et al., 2008*).

Beyond regulating synthesis and assembly of FN, which orchestrates the deposition of major basement membrane components (i.e. collagen IV and perlecan) that are required for stability and maturation of this specialized ECM (*Botta et al., 2012*; *Marchand et al., 2019*), our data revealed a broader role for SAM68 in matrix conditioning. Notably, we found that SAM68 regulates mRNA expression of basement membrane-associated components including collagen VIII, the glycoproteins periostin and fibulin-1, and the proteoglycan biglycan. Type VIII collagen is a non-fibrillar collagen associated with vascular basement membranes for which a role in endothelial cell sprouting in vitro has been described (*Sage and Iruela-Arispe, 1990*). Periostin is an ECM scaffolding protein with multiple binding sites for FN and collagens (*Kii, 2019*). Thus, decreased periostin expression by endothelial cells could exacerbate the defect in FN-dependent collagen assembly. Incorporation of fibulin-1 in the subendothelial matrix is crucial for vessel formation during embryonic development (*Ito et al., 2020*; *Kostka et al., 2001*). Interestingly, in the light of our data, the FN-interacting proteoglycan biglycan is induced after wounding where it selectively associates with lamellopodia at the leading edge of migrating cells (*Kinsella et al., 1997*). Thus, our results place SAM68 as a key player in shaping and conditioning the perivascular matrix, which provides an important source of mechanical and angiocrine signals, through embedded growth and angiogenic factors.

Sprouting angiogenesis involves pro-invasive endothelial tip cells that extend numerous filopodial protrusions and cohesive stack cells that ensure connections to the primary vessel. Our fibrin gel sprouting assay revealed that SAM68-deficient endothelial cells produce shorter capillary-like structures, as compared to control cells, with tip cells that form stunted protrusive structures and frequently detach from the stalks. These results suggest that both adhesion-stabilizing and ECM gene-regulatory functions of SAM68 are required for capillary-like formation, however we cannot exclude that additional functions of the RBP may be involved in this 3D setting.

Finally, beyond the functions described above, it is tempting to speculate that SAM68 plays a role in integrin-dependent phagocytic processes in endothelial cells, which are implicated in clearance of pathogens, apoptotic cells and ECM debris during tissue remodeling. Such processes are investigated using FN-coated beads and utilize the same molecular machinery as integrin-mediated firm adhesion, as reviewed in (*Dupuy and Caron, 2008*).

In sum, our study illustrates a striking example of the multifaceted functions of RBPs in dynamic adhesion and actin-dependent membrane remodeling processes. With a focus on SAM68 in endothelial cells, we described the dynamic localization and functions of this adhesome-associated RBP at integrin adhesion sites and in the nucleus. Our findings should provide impetus for future studies to further decipher the mechanisms underlying RBP-dependent regulation of angiogenic processes.

## Materials and methods

### Materials

Plasma FN was from Corning (Bedford, MA, USA, #356008) Alexa Fluor 488- or 647-conjugated phalloidin and Hoechst 33342 were purchased from Invitrogen (Eugene, Oregon, USA, #A12379, #A22287, #H3570). Reagents for the 3D sprouting angiogenesis assay (Fibrinogen/#F8630, Thrombin/#T4648, Aprotinin/#A1153) were purchased from Sigma-Aldrich (Saint-Louis, MI, USA). Antibodies and oligonucleotides used in this study are indicated in the **Key Resources Table** (**Appendix 1**). All raw data associated with this manuscript are available in the source data files. All materials generated in this study are available upon request.

### Cells and culture conditions

Primary human umbilical vein endothelial cells (HUVECs) were prepared from fresh human umbilical veins as previously described (*Barbieri et al., 1981*) and maintained in Human Endothelial SFM (Gibco, Life Technologies Corporation, Grand Island, NY, USA, #11111044) supplemented with 20%

Fetal Bovine Serum (FBS) (Dutsher, Bernolsheim, France, #500105 A1A), epidermal growth factor (EGF, 10 ng/mL, Invitrogen), bFGF (10 ng/ml, prepared in the protein purification facility of our institute), heparin (10 ng/ml, Sigma, Saint Quentin Fallavier, France, #H3149), and antibiotics (Gibco, #15140–122). HUVECs were used up to the 6th passage for all experiments. For assessment of FN protein production and deposition, cells were trypsinized (Gibco, #15400–054) and grown in 2% FN-depleted serum for 48 hr before protein extraction. The HEK293FT cell line from Life Technologies (Saint Aubin, France) was maintained in DMEM (Gibco, #31966–021) supplemented with 10% FBS and 1 X Non-essential Amino Acid supplement (Sigma, #M7145). Absence of Mycoplasma sp. contamination was routinely verified by PCR as described elsewhere (*Kong et al., 2001*).

## Coating and substrates for cell adhesion
Where indicated, culture plates or glass coverslips were coated with 10 µg/ml plasma FN in PBS for 1 hr at 37 °C. After coating, plates were washed in PBS and air-dried under the culture hood for 10 min.

## Plasmid constructs
To generate the *FN1* promoter activity reporter construct, 3500 nucleotides located upstream of the transcription initiation of FN mRNA were amplify from HUVEC genomic DNA using KAPA HiFi HotStart (Kapa Biosystems, Wilmington, MA, USA, #07958889001) and cloned upstream of the luciferase coding sequence of the pGL3-Basic plasmid (Promega, Madison, WI, USA, # E1751). The SAM68 expression plasmid was generated by insertion of the complete SAM68 coding sequence, amplified from HUVEC cDNA and cloned in the pcDNA3.1 plasmid (Sigma-Aldrich, #E0648). The SAM68 WT lentivirus expression vector was generated by insertion of a FLAG-tagged SAM68 coding sequence into pLenti-CMV-MCS-GFP-SV-puro plasmid (gift from Paul Odgren Addgene plasmid # 73582) between XbaI and MluI restriction sites. The SAM68 P358A mutant was generated by site-directed mutagenesis and the SAM68 ΔKH mutant was created by removal of the sequences encoding residues 157–256 from the cDNA fragment. All constructs were checked by full-length sequencing (Microsynth AG, Balgach, Switzerland).

## Methods siRNA and blocking oligonucleotide transfection
Primary endothelial cells were transfected in Opti-MEM (Gibco, #51985–026) with either scrambled siRNA or 2 different siRNAs targeting human SAM68 mRNA sequence at a final concentration of 20 nM using Lipofectamine RNAiMAX (Invitrogen, #13778). Sequence design of SAM68 siRNAs was performed using the Web-based DSIR tool (*Filhol et al., 2012*). One day following transfection, cells were trypsinized and plated as indicated. The same procedure has been followed for blocking oligonucleotide transfection at a final concentration of 500 nM consisting of an equimolar mix of both blocking oligos (#SBE1 and #SBE2) or scrambled (CTRL) oligos using RNAiMAX. Successful transfection was assessed by live imaging of CTRL-Cy3 labeled scrambled oligos.

## Lentivirus production and transduction
Lentiviral particles were generated according to standard Addgene protocol (https://www.addgene.org/protocols/lentivirus-production/) in the HEK293FT packaging cell line. Viral particles were harvested after 36 hr of production and used to infect 10⁶ low passage endothelial cells the same day.

## Protein extraction and western blot analyses
Cells were lysed in 3 X Laemmli Buffer and protein quantification was performed using the Pierce BCA protein assay kit (Thermo Scientific, Rockford, IL, USA #23225). Proteins were separated on 7.5% SDS–PAGE gels and transferred to Amersham Hybond PVDF membranes (GE Healthcare, Chicago, IL, USA, #10600023) overnight at 4 °C (30 Volts) in Dunn buffer. Membranes were saturated in PBS 5% non-fat dry milk for 1 hr at RT and incubated in PBS 3% BSA with primary antibody (listed in Appendix 1—key resources table) overnight at 4 °C. Secondary antibodies conjugated to Horseradish Peroxidase were incubated in PBS 5% non-fat dry milk for 1 hr at RT. Immunostained bands were detected using Clarity Western ECL (Biorad, Hercules, CA, USA, # 170–5060). Densitometric analysis of the band were performed using FIJI software (*Schindelin et al., 2012*).

## Luciferase reporter assay

$2.5 \times 10^5$ HEK293 cells were plated in 12-well plates, transfected using CaCl2 (*Chen, 2012*) with 0.25 μg of pFN-Luc and indicated amounts of either the SAM68 expression plasmid (pcDNA3.1-SAM68) or pcDNA3.1 empty vector and 25 ng of plasmid encoding Renilla luciferase under the control of thymidine kinase promoter (pRL, Promega, #E2231) for normalization. The next day, cells were serum starved for 4 hr then treated for 6 hr with either DMSO or 40 ng of TPA (Sigma-Aldrich, Saint-Louis, M, USA, #P8139). Firefly and renilla luciferase activities were measured sequentially with the Dual Luciferase Reporter Assay (Promega, Madison, WI, USA, #E1910) using a Centro LB 960, BERTHOLD Technologies luminometer. Results are expressed as relative light units of firefly luciferase activity over relative light units of Renilla luciferase activity.

## SAM68 ChIP assay

RNP complexes were immunoprecipitated according to Abcam protocols (Cambridge, CB2 0 AX, UK). Briefly, endothelial cells were grown to 80% confluency, incubated in 1% formaldehyde for 10 min at room temperature and the reaction was stopped with 0.25 M glycine. After scrapping, cells were resuspended and lysed in 50 mM HEPES, 140 mM NaCl, 1 mM EDTA and 0.5% Triton X-100 for 30 min on ice. Cell lysates were sonicated (Bioruptor Diagenode Sonicator, Diagenode, Seraing, Belgium) and immunoprecipitated using Magnetic Dynabeads (Invitrogen, #112030) preincubated with nonimmune IgG (Invitrogen, #31235) or anti-SAM68 antibody (Sigma, #HPA051280). After reversion of the cross-link, immunoprecipitated complexes were treated with RNase A and proteinase K. DNA fragments were recovered by phenol-chloroform extraction followed by ethanol precipitation and assayed by qPCR to determine the fold enrichment of *FN1* promoter sequences in SAM68 immunoprecipitated complexes, compared to nonimmune IgG complexes. Statistical analysis of fold enrichment was performed with R using Pairwise T-test with p-values adjusted using 'Bonferroni correction'.

## RNA extraction and gene expression analysis by quantitative real-time PCR amplification

RNA extraction was performed using the RNA XS extraction kit (Macherey-Nagel GmbH & Co. KG, Düren, Germany, #740902) and cDNAs were generated from 1 μg of RNA using a High-Capacity cDNA Reverse Transcription kit (Applied Biosystems, Waltham, MA, USA, # 10400745). qPCR was performed on a StepOnePlus System (Applied Biosystems) using the PowerUp SYBR Green Master Mix (Applied Biosystems, #A25742) and relative expression levels were calculated using the ΔΔCt method, normalized to RPL27a mRNA levels unless specified.

## Cell detachment assay

$2.5 \times 10^5$ siRNA-transfected HUVECs were plated in 6-well plates and grown for 48 hr. Cells were gently detached by adding 1 ml of a 1:150 diluted trypsin solution (Gibco, #15400–054) for exactly 2 min at 37 °C. Fractions of detached cells were collected and 1 ml of fresh diluted trypsin was added to the wells. Twelve cycles of trypsinization were performed (no washing between steps) until all cells were detached. Cells collected in each fraction were counted and expressed as a percentage of total collected cells.

## Ectopic adhesion complex formation assay

A total of $5.10^8$ particles/ml of Polybead Microspheres (Polysciences, Warrington, PA, USA, #17135–5-4.5 μm) were coated overnight at 4 °C with plasma FN (10 μg/ml) in PBS. Microbeads were then deposited directly on endothelial cells (30 beads per cell) which had been plated overnight on uncoated glass coverslips and serum-starved for 1 hr prior to bead addition. Cells were fixed for analysis 20 min after bead addition.

## 2D cell migration assays

For wound healing assays, confluent monolayers of siCTRL- or siSAM68-transfected HUVECs were manually scratched using a plastic tip, rinsed with warm PBS and maintained in complete medium supplemented with 2% FBS and 20 mM HEPES (Gibco, #15630–056). Video microscopy was performed as previously described (*Radwanska et al., 2017*) using a 10 x air objective (NA 0.25) on a Zeiss Axiovert 200 M microscope equipped with a sCMOS NEO camera (Andor, Belfast, UK). Image acquisition at 10 min intervals was performed using the MetaMorph Imaging System (Universal Imaging

Corp., Downingtown, PA). Scratch width and surface over time were extracted from films using the Wound Healing Size plugin for ImageJ (*Suarez-Arnedo et al., 2020*). For sparse cell migration analyses, SAM68-depleted cells or control cells were plated overnight in complete medium, incubated for 30 min with fluorescent UE1 lectin (Sigma-Aldrich, #L9006) and washed twice with warm PBS before live imaging. Transmission and fluorescent images of lectin were acquired on the same Zeiss Axiovert 200 M microscope at 10 min intervals and cell displacement was tracked using the TrackMate plugin of FIJI (*Tinevez et al., 2017*). Statistical analysis of migrating cells was performed on cells tracked between 2 hr (equilibration in the chamber) and 8 hr.

### 3D in vitro sprouting angiogenesis test

Sprouting of siRNA transfected cells was evaluated according to the procedure described in *Kempers et al., 2021*. Briefly, Cytodex 3 microcarrier beads (Cytiva Sweden AB, Uppsala, Sweden, #17048501) were coated with endothelial cells and embedded in a fibrin gel for 48 hr in complete culture medium. Cells were then fixed in 4% paraformaldehyde (EMS, Electron Microscopy Sciences, Hatfield, PA, USA, #15713)/ 5% sucrose at 37 °C and nuclei and F-actin were stained with Hoechst 33342 and Alexa Fluor 488- or 647-conjugated phalloidin, respectively, for imaging using a Zeiss NLO780 confocal microscope with a 20 X objective (NA 0.8).

### Immunostaining

Cells grown on 18- or 25 mm round glass slides (EMS, #EM-72290–12) were fixed in 4% paraformaldehyde / 5% sucrose at 37 °C, washed in PBS and permeabilized in PBS 0.5% Triton. Blocking of unspecific interactions was performed in PBS containing 4% BSA (Euromedex, Souffelweyersheim, France, 04-100-812-C) for 1 hr at RT followed by overnight 4 °C staining with primary antibodies diluted in blocking solution. Cells were then washed thoroughly with blocking solution and incubated with respective secondary antibodies diluted 1/4000 in blocking solution at RT for 1 hr. A final step of extensive washes in PBS is performed before air drying of the samples and mounting in Prolong Gold antifade (Invitrogen P36930). Cells imaged using TIRF confocal microscopy were not mounted and left at °4 C in PBS.

### β-actin mRNA in-situ hybridization (RNA-FISH)

Following ectopic adhesion formation, cells were fixed for 20 min in a 4% PFA/ 5% sucrose solution then washed 3 X in PBS. Thereafter, fixed cells were permeabilized in 70% ethanol at 4 °C overnight. The next day, β-actin-FLAP-Cy3-coupled oligonucleotide probes (see primer list) were hybridized to endogenous β-actin mRNA according to the smiFISH protocol (*Tsanov et al., 2016*), using Stellaris Buffer (LGC Biosearch Technologies, Hoddesdon, United Kingdom, #SMF-HB1-10) for all the hybridation steps. Probes were designed using the Oligostan R script available at: https://bitbucket.org/muellerflorian/fish_quant (copy archived at *Mueller, 2023*) and are listed in Appendix 1—key resources table.

### Cell imaging and microscopy

Unless otherwise specified, confocal imaging was performed on a Nikon ECLIPSE Ti 100 x (SPINNING DISC) using Metamorph Acquisition software (Molecular Devices, San Jose, CA, USA) using 40 X water 1.2 NA, or 100 X oil objective (NA 1.4) with the LASERs 405, 488, 561, 633 nm. For live cell imaging of eGFP-SAM68 /LifeAct Ruby sequence of 30–60 s duration were acquired on TIRF illumination mode every 100ms. Whole coverslip imaging of the ECM, was carried out using the PhenoImager HT (formerly Vectra Polaris) slide scanner in the 40 X mode (Akoya Biosciences, Marlborough, MA, USA).

### Statistical analysis and graphs

Statistical analyses were performed with either R (*R Development Core Team, 2021*) or Prism 8 (GraphPad Software, San Diego, CA, USA) software.

The paired Student's t-test was applied, unless specified in the legend. Differences were considered to be statistically significant at $p<0.05$ (*), $p<0.01$ (**), $p<0.001$ (***) and data are expressed as mean ± SD. N=number of biologically distinct experiments; n=number of observations per condition.

### Image analysis methods and workflow

See *Figure 9* for image analysis workflow.

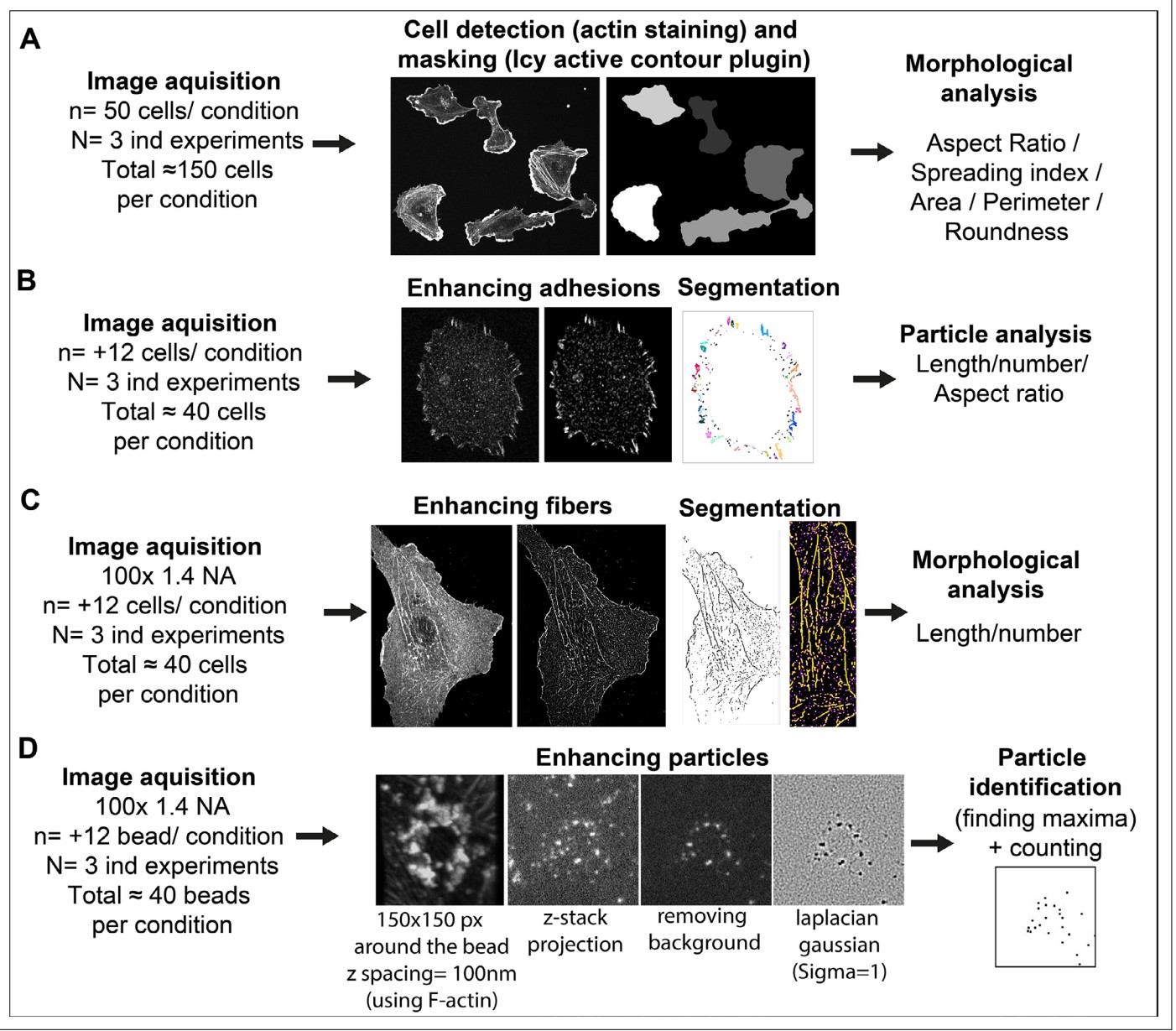

**Figure 9.** Image analysis workflow. (**A**) *Cell Morphology*. General morphology of cells grown overnight on glass coverslips was assessed using F-actin staining. A mask of each individual cell was created with the 'active contour' plugin (*Dufour et al., 2011*) of ICY software (*de Chaumont et al., 2012*). Resulting binary masks were then imported in FIJI (*Schindelin et al., 2012*) for subsequent morphological feature extraction using the MorphoLibJ plugin (*Legland et al., 2016*). (**B**) *Adhesion site analysis*. Overnight HUVEC cultures were stained for vinculin and a z-stack (5 µm from coverslip contact site) was imaged. Intensities, brightness, and contrast of the focal plane was modified to enhance adhesive structures. Resulting images were then thresholded and diffuse vinculin staining in the masks was removed with a 5 µm ellipse circling the nucleus. Resulting segmented adhesions were quantified and described using the FIJI particle analysis plugin. Stained structures under 0.25 µm were not considered as focal complexes (*Geiger et al., 2001*). (**C**) *Fibrillar adhesion analysis*. Intensities, brightness, and contrast of the focus plan was modified to enhance these structures. Resulted images were then segmented using classic thresholding methods, and further skeletonized; fibrillar adhesions were analyzed using the FIJI analyze skeleton plugin. (**D**) *pFAK-Y397 and β-actin mRNA particle analysis*. Five µm z-stacks (100 nm spacing) were acquired around each FN-coated bead engaged in adhesion formation near the cell periphery. Cell attachment to each bead was assessed using F-actin staining when forming a dense 'cup' like ring structure around the bead. Background noise was removed (Fiji Process) and particles surrounding the beads (150-pixel square area) were enhanced using Laplacian gaussian filter (sigma = 1) followed by maxima detection using 'FeatureJ' plugin. For calculation of specific β-actin particle enrichment at the bead, an additional area of 150x150pixel from apical to basal plane of the cell was selected in the cytoplasm (non-bead area) of the same cell for quantification, as reference.

## Matrix deposition

Analysis of ECM deposition was performed on whole-coverslip scanned images with Halo bioimage analysis software (Indica Labs, Albuquerque, NM, USA). Briefly, the area corresponding to the fluorescent signal of each matrix component was segmented using the 'Area Quantification FL' plugin of the software and subsequent measurements (in µm²) were normalized to the number of cells on each slide. Number of cells was assessed by nuclei count after segmentation using the nuclei segmentation plugin of the software.

## Acknowledgements

We gratefully acknowledge Sameh Ben Aicha and Baptiste Monterroso (iBV PRISM Imaging facility), and Samah Rekima (iBV Histopathology facility) for expert assistance with microscopy and image analysis. Arnaud Hubstenberger and Florence Besse are kindly acknowledged for insightful discussions and help with smiRNA-FISH experiments. We are also grateful to Laurent Gagnoux-Palacios for helpful discussions.

Support for this work was provided by the National Agency for Research (ANR-16-CE93-0005-01 and ANR-19-P3IA-0002 (3IA Côte d'Azur)), the LabEx SIGNALIFE program (ANR-11-LABX-0028–01), the Canceropôle Provence Alpes Côte d'Azur (Emergence program 2022), Institut National du Cancer and Région Sud and the Université Côte d'Azur (Crédits Scientifiques Incitatifs-2022).

## Additional information

### Funding

| Funder | Grant reference number | Author |
| --- | --- | --- |
| Agence Nationale de la Recherche | ANR-16-CE93-0005-01 | Ellen Van Obberghen-Schilling |
| Agence Nationale de la Recherche | ANR-19-P3IA-0002 (3IA Côte d'Azur) | Ellen Van Obberghen-Schilling |
| Agence Nationale de la Recherche | ANR-11-LABX-0028-01 | Ellen Van Obberghen-Schilling |
| Canceropôle PACA | Emergence program 2022 | Delphine Ciais |
| Université Côte d'Azur | CSI 2022 | Delphine Ciais |

The funders had no role in study design, data collection and interpretation, or the decision to submit the work for publication.

### Author contributions

Zeinab Rekad, Conceptualization, Resources, Data curation, Formal analysis, Validation, Investigation, Methodology, Writing – review and editing; Michaël Ruff, Dominique Grall, Validation, Investigation, Methodology; Agata Radwanska, Resources, Validation, Methodology; Delphine Ciais, Conceptualization, Resources, Data curation, Formal analysis, Supervision, Funding acquisition, Validation, Investigation, Methodology, Writing – original draft, Project administration, Writing – review and editing; Ellen Van Obberghen-Schilling, Conceptualization, Resources, Formal analysis, Supervision, Funding acquisition, Methodology, Project administration, Writing – review and editing

### Author ORCIDs

Zeinab Rekad ⓘ http://orcid.org/0000-0002-1946-0483
Delphine Ciais ⓘ https://orcid.org/0000-0002-1649-897X
Ellen Van Obberghen-Schilling ⓘ http://orcid.org/0000-0003-2961-0059

### Decision letter and Author response

Decision letter https://doi.org/10.7554/eLife.85165.sa1
Author response https://doi.org/10.7554/eLife.85165.sa2

# Additional files

## Supplementary files
• MDAR checklist

## Data availability
All raw data associated with this manuscript are available in the source data files. All materials generated in this study are available upon request.

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

# Appendix 1

## Appendix 1—key resources table

| Reagent type (species) or resource | Designation | Source or reference | Identifiers | Additional information |
|---|---|---|---|---|
| Antibody | anti-Vinculin (mouse monoclonal) | Sigma-Aldrich | Cat# V 9131, RRID:AB_477629 | IF (1:200) |
| Antibody | anti-SAM68 (rabbit polyclonal) | Sigma-Aldrich | Cat # HPA051280 | IF (1:200) WB (1:200) |
| Antibody | anti-Phosphotyrosine (rabbit polyclonal) | Sigma-Aldrich | Cat #05–321, RRID:AB_2891016 | IF (1:200) |
| Antibody | anti-FAK (rabbit polyclonal) | Cell Signaling | Cat# 3285 | IF (1:200) |
| Antibody | anti-pFAK-Y397(rabbit polyclonal) | Cell Signaling (#3283) | Cat# 3283 | IF (1:200) |
| Antibody | anti-Integrin α5β1 (mouse monoclonal) | Chemicon international | Cat# MAB1999 | IF (1:100) |
| Antibody | anti-Fibronectin (mouse monoclonal) | BD Transduction Laboratories | Cat# 610077 | IF (1:500) WB (1:4000) |
| Antibody | anti-HSP90 (mouse monoclonal) | Thermo Fisher Scientific | Cat# MA1-10372, RRID:AB_11155433 | WB (1:1000) |
| Antibody | anti-Perlecan (rat monoclonal) | Thermo Fisher Scientific | Cat# RT-794 | IF (1:200) |
| Antibody | anti-Collagen IV | Novotec | Cat# 20411 | IF (1:100) |
| Antibody | anti-mouse Alexa Fluor 488 (goat polyclonal) | Invitrogen | Cat# A-11029 | IF (1:4000) |
| Antibody | anti-mouse Alexa Fluor 546 (goat polyclonal) | Invitrogen | Cat# A-11030 | IF (1:4000) |
| Antibody | anti-rabbit Alexa Fluor 488 (goat polyclonal) | Invitrogen | Cat# A-11034 | IF (1:4000) |
| Antibody | anti-rabbit Alexa Fluor 546 (goat polyclonal) | Invitrogen | Cat# A-11035 | IF (1:4000) |
| Antibody | anti-rat Alexa Fluor 488 (goat polyclonal) | Invitrogen | Cat# A-11006 | IF (1:4000) |
| Antibody | anti-rat Alexa Fluor 546 (goat polyclonal) | Invitrogen | Cat# A-11081 | IF (1:4000) |
| Sequence-based reagent | Control siRNA (siCTL) | Eurogentec, Seraing, Belgium | Cat# SR-CL000-005 | nontargeting siRNA |
| Sequence-based reagent | SAM68 siRNA (si68-1)_F | Eurogentec, Seraing, Belgium | siRNA | CAGGAUUCCUGUUGCUUUACC |
| Sequence-based reagent | SAM68 siRNA (si68-1)_R | Eurogentec, Seraing, Belgium | siRNA | UAAAGCAACAGGAAUCCUGGG |
| Sequence-based reagent | SAM68 siRNA (si68-2)_F | Eurogentec, Seraing, Belgium | siRNA | GGAAGUCAAGAAAUUUCUAGU |
| Sequence-based reagent | SAM68 siRNA (si68-2)_R | Eurogentec, Seraing, Belgium | siRNA | UAGAAAUUUCUUGACUUCCUC |
| Sequence-based reagent | RPL27a_F | Eurogentec, Seraing, Belgium | qPCR primer | AGAGCTTCTGCCCAACTGTC |
| Sequence-based reagent | RPL27a_R | Eurogentec, Seraing, Belgium | qPCR primer | TCACGATGACAGGCTGCTTT |
| Sequence-based reagent | t-FN qPCR_F | Eurogentec, Seraing, Belgium | qPCR primer (all FN variants) | GGGTCATGTACCGCATTGGA |
| Sequence-based reagent | t-FN qPCR_R | Eurogentec, Seraing, Belgium | qPCR primer (all FN variants) | GACGCTTGTGGAATGTGTCG |
| Sequence-based reagent | EDA-FN qPCR_F | Eurogentec, Seraing, Belgium | qPCR primer | TGAGCTATTCCCTGCACCTG |
| Sequence-based reagent | EDA-FN qPCR_R | Eurogentec, Seraing, Belgium | qPCR primer | GTGGGTGTGACCTGAGTGAA |

*Appendix 1 Continued on next page*

*Appendix 1 Continued*

| Reagent type (species) or resource | Designation | Source or reference | Identifiers | Additional information |
|---|---|---|---|---|
| Sequence-based reagent | EDB-FN qPCR_F | Eurogentec, Seraing, Belgium | qPCR primer | TGGTCCATGCTGATCAGAGC |
| Sequence-based reagent | EDB-FN qPCR_R | Eurogentec, Seraing, Belgium | qPCR primer | CCTCAGGCCGATGCTTGAAT |
| Sequence-based reagent | COL4A1_F | Eurogentec, Seraing, Belgium | qPCR primer | GGCCAGAAAGGAGAGATGGG |
| Sequence-based reagent | COL4A1_R | Eurogentec, Seraing, Belgium | qPCR primer | ATCAACAGATGGGGTGCCTG |
| Sequence-based reagent | COL8A1_F | Eurogentec, Seraing, Belgium | qPCR primer | CAAGGGAGCTCACACGTTCA |
| Sequence-based reagent | COL8A1_R | Eurogentec, Seraing, Belgium | qPCR primer | GGGGCTGGTTTCTGTCTCTT |
| Sequence-based reagent | LAMA4_F | Eurogentec, Seraing, Belgium | qPCR primer | GAAGACATGAACAGGGCCAC |
| Sequence-based reagent | LAMA4_R | Eurogentec, Seraing, Belgium | qPCR primer | GAGGTGTTGTCAGAGAGTCCG |
| Sequence-based reagent | NID1_F | Eurogentec, Seraing, Belgium | qPCR primer | GTGTGGAGGGCTACCAGTTT |
| Sequence-based reagent | NID1_R | Eurogentec, Seraing, Belgium | qPCR primer | GCTGGGGTATGTCGCAGTTA |
| Sequence-based reagent | HSPG2_F | Eurogentec, Seraing, Belgium | qPCR primer | TGCGCTGGACACATTCGTA |
| Sequence-based reagent | HSPG2_R | Eurogentec, Seraing, Belgium | qPCR primer | ACTCGATGGAGCGAGTGAAAT |
| Sequence-based reagent | POSTN_F | Eurogentec, Seraing, Belgium | qPCR primer | AAGGAATGAAAGGCTGCCCA |
| Sequence-based reagent | POSTN_R | Eurogentec, Seraing, Belgium | qPCR primer | GTCAGAATAGCGCTGCGTTG |
| Sequence-based reagent | FBLN1_F | Eurogentec, Seraing, Belgium | qPCR primer | CTTCCGGCTCTCTGTGGATG |
| Sequence-based reagent | FBLN1_R | Eurogentec, Seraing, Belgium | qPCR primer | ACACTGGTAGGAGCCGTAGA |
| Sequence-based reagent | BGN_F | Eurogentec, Seraing, Belgium | qPCR primer | GCCAACTAGTCAGCCTGCG |
| Sequence-based reagent | BGN_R | Eurogentec, Seraing, Belgium | qPCR primer | CCATCGTCCAGGGTGAAGTC |
| Sequence-based reagent | 18 S rRNA_F | Eurogentec, Seraing, Belgium | qPCR primer | CGGCGACGACCCATTCGAAC |
| Sequence-based reagent | 18 S rRNA_R | Eurogentec, Seraing, Belgium | qPCR primer | GAATCGAACCCTGATTCCCCGTC |
| Sequence-based reagent | FN Promotor (–600 fragment)_F | Eurogentec, Seraing, Belgium | qPCR primer | GAAGAAGTCCGAACAGGGAGCTGTG |
| Sequence-based reagent | FN Promotor (–600 fragment)_R | Eurogentec, Seraing, Belgium | qPCR primer | GGCTGCCTTTCCCCCCATCCCGCTC |
| Sequence-based reagent | FN Promotor (–1000 fragment)_F | Eurogentec, Seraing, Belgium | qPCR primer | GCGGGGGATGGAGGGGGCATTCTGT |
| Sequence-based reagent | FN Promotor (–1000 fragment)_R | Eurogentec, Seraing, Belgium | qPCR primer | TATGTACTGTCTTGCCCTCCTTCGG |
| Sequence-based reagent | FN Promotor (–1500 fragment)_F | Eurogentec, Seraing, Belgium | qPCR primer | TGGGTCACAAAGATTCCTCAAGAGG |
| Sequence-based reagent | FN Promotor (–1500 fragment)_R | Eurogentec, Seraing, Belgium | qPCR primer | CAAGGATTTAAAACCAAACCAAAAC |
| Sequence-based reagent | FN Promotor (–3000 fragment)_F | Eurogentec, Seraing, Belgium | qPCR primer | GTTCTGTCTCTACCACATATATGCC |

*Appendix 1 Continued on next page*

*Appendix 1 Continued*

| Reagent type (species) or resource | Designation | Source or reference | Identifiers | Additional information |
|---|---|---|---|---|
| Sequence-based reagent | FN Promotor (–3000 fragment)_R | Eurogentec, Seraing, Belgium | qPCR primer | GACTTGCTCTCAGGTAGCAGCAAC |
| Sequence-based reagent | CTL antisense blocking oligonucleotide | Eurogentec, Seraing, Belgium | blocking oligonucleotide | (2'Ome)UCG-T(2'Ome)CA-CC(2'Ome)A-ATG-(2'Ome)CGT-T(2'Ome)AA-TG(2'Ome)U |
| Sequence-based reagent | SBE1 antisense blocking oligonucleotide | Eurogentec, Seraing, Belgium | blocking oligonucleotide | (2'Ome)AGT-G(2'Ome)AC-TA(2'Ome)C-TAA-(2'Ome)AAA-A(2'Ome)AC-CA(2'Ome)A-A |
| Sequence-based reagent | SBE2 antisense blocking oligonucleotide | Eurogentec, Seraing, Belgium | blocking oligonucleotide | (2'Ome)AAA-C(2'Ome)AA-TG(2'Ome)U-ACA-(2'Ome)ATC-A(2'Ome)AA-GT(2'Ome)C-C |
| Sequence-based reagent | hACTB-P1 | Eurogentec, Seraing, Belgium | smRNA FISH oligonucleotide | AAGGTGTGCACTTTTATTCAACTGGTCTCAAGCCTCCTAAGTTTCGAGCTGGACTCAGTG |
| Sequence-based reagent | hACTB-P2 | Eurogentec, Seraing, Belgium | smRNA FISH oligonucleotide | AGAAGCATTTGCGGTGGACGATGGAGGGGCCTCCTAAGTTTCGAGCTGGACTCAGTG |
| Sequence-based reagent | hACTB-P3 | Eurogentec, Seraing, Belgium | smRNA FISH oligonucleotide | GCTCAGGAGGAGCAATGATCTTGATCTTCCCTCCTAAGTTTCGAGCTGGACTCAGTG |
| Sequence-based reagent | hACTB-P4 | Eurogentec, Seraing, Belgium | smRNA FISH oligonucleotide | GGATGTCCACGTCACACTTCATGATGGAGCCTCCTAAGTTTCGAGCTGGACTCAGTG |
| Sequence-based reagent | hACTB-P5 | Eurogentec, Seraing, Belgium | smRNA FISH oligonucleotide | GAAGGTAGTTTCGTGGATGCCACAGGACTCCCTCCTAAGTTTCGAGCTGGACTCAGTG |
| Sequence-based reagent | hACTB-P6 | Eurogentec, Seraing, Belgium | smRNA FISH oligonucleotide | CAGCGGAACCGCTCATTGCCAATGGTCCTCCTAAGTTTCGAGCTGGACTCAGTG |
| Sequence-based reagent | hACTB-P7 | Eurogentec, Seraing, Belgium | smRNA FISH oligonucleotide | CAGCCTGGATAGCAACGTACATGGCTGCCTCCTAAGTTTCGAGCTGGACTCAGTG |
| Sequence-based reagent | hACTB-P8 | Eurogentec, Seraing, Belgium | smRNA FISH oligonucleotide | GTGTTGAAGGTCTCAAACATGATCTGGGTCATCCTCCTAAGTTTCGAGCTGGACTCAGTG |
| Sequence-based reagent | hACTB-P9 | Eurogentec, Seraing, Belgium | smRNA FISH oligonucleotide | TCGGGAGCCACACGCAGCTCATTGTACCTCCTAAGTTTCGAGCTGGACTCAGTG |
| Sequence-based reagent | hACTB-P10 | Eurogentec, Seraing, Belgium | smRNA FISH oligonucleotide | ACGAGCGCGGCGATATCATCATCCATCCTCCTAAGTTTCGAGCTGGACTCAGTG |
| Sequence-based reagent | hACTB-P11 | Eurogentec, Seraing, Belgium | smRNA FISH oligonucleotide | TTCTCCTTAGAGAGAAGTGGGGTGGCTTTTAGCCTCCTAAGTTTCGAGCTGGACTCAGTG |
| Sequence-based reagent | hACTB-P12 | Eurogentec, Seraing, Belgium | smRNA FISH oligonucleotide | CATTGTGAACTTTGGGGGATGCTCGCTCCCTCCTAAGTTTCGAGCTGGACTCAGTG |
| Sequence-based reagent | hACTB-P13 | Eurogentec, Seraing, Belgium | smRNA FISH oligonucleotide | GACTGCTGTCACCTTCACCGTTCCAGCCTCCTAAGTTTCGAGCTGGACTCAGTG |
| Sequence-based reagent | hACTB-P14 | Eurogentec, Seraing, Belgium | smRNA FISH oligonucleotide | GGACTCGTCATACTCCTGCTTGCTGACCTCCTAAGTTTCGAGCTGGACTCAGTG |
| Sequence-based reagent | hACTB-P15 | Eurogentec, Seraing, Belgium | smRNA FISH oligonucleotide | CAGTGATCTCCTTCTGCATCCTGTCGCCTCCTAAGTTTCGAGCTGGACTCAGTG |
| Sequence-based reagent | hACTB-P16 | Eurogentec, Seraing, Belgium | smRNA FISH oligonucleotide | GACAGCACTGTGTTGGCGTACAGGTCTTTCCTCCTAAGTTTCGAGCTGGACTCAGTG |
| Sequence-based reagent | hACTB-P17 | Eurogentec, Seraing, Belgium | smRNA FISH oligonucleotide | CGTGGCCATCTCTTGCTCGAAGTCCACCTCCTAAGTTTCGAGCTGGACTCAGTG |
| Sequence-based reagent | hACTB-P18 | Eurogentec, Seraing, Belgium | smRNA FISH oligonucleotide | GCGACGTAGCACAGCTTCTCCTTAATGTCCTCCTAAGTTTCGAGCTGGACTCAGTG |
| Sequence-based reagent | hACTB-P19 | Eurogentec, Seraing, Belgium | smRNA FISH oligonucleotide | AGGTGTGGTGCCAGATTTTCTCCATGTCGCCTCCTAAGTTTCGAGCTGGACTCAGTG |
| Sequence-based reagent | hACTB-P20 | Eurogentec, Seraing, Belgium | smRNA FISH oligonucleotide | CCAGTTGGTGACGATGCCGTGCTCGATCCTCCTAAGTTTCGAGCTGGACTCAGTG |
| Sequence-based reagent | hACTB-P21 | Eurogentec, Seraing, Belgium | smRNA FISH oligonucleotide | GGTACTTCAGGGTGAGGATGCCTCTCTCCTCCTAAGTTTCGAGCTGGACTCAGTG |
| Sequence-based reagent | hACTB-P22 | Eurogentec, Seraing, Belgium | smRNA FISH oligonucleotide | CCTCGTCGCCCACATAGGAATCCTTCCCTCCTAAGTTTCGAGCTGGACTCAGTG |

