## [Editor Report]

This paper provides important evidence that the RNA binding protein SAM68 regulates endothelial cell migration through multiple mechanisms including localizing actin mRNA to focal adhesions and stimulating transcription of the fibronectin gene. The evidence is generally convincing, although the relative roles of transcription and RNA localization in SAM68 functions and the dynamics of RNA movement to adhesion sites remain unknown. The paper will be of interest to cell biologists investigating post-transcriptional regulatory mechanisms.

---

## [Decision Letter]

[Editors' note: this paper was reviewed by Review Commons.]

---

## [Author Response]

General Statements [optional]

The goal of our study was to evaluate the role of the RNA binding protein SAM68 in the regulation of cell adhesion and adaptation of endothelial cells to their extracellular environment. We showed that SAM68 depletion affected endothelial cell behavior by impairing adhesion site maturation and compromising basement membrane assembly.

We are pleased that the reviewers found our study to be interesting, well written and clear, with findings that are supported by carefully designed experiments. Importantly, we would like to thank the reviewers for their careful analysis of our work and for their clear and constructive comments.

One common query was whether the regulation of β-actin mRNA localization at adhesion sites, and *FN1* gene transcription by SAM68 in endothelial cells involves direct interactions with the mRNA and promoter, respectively. This important point will be addressed with additional experiments in order to strengthen our hypothesis.

A second point that emerged from the reviews relates to the interdependence of SAM68 multi-layered effects on cell adhesions and *FN1* gene transcription. Our response to this issue is discussed below and has been clarified in the revised manuscript.

Lastly, since in vivo studies are not feasible locally or in a reasonable timeframe, our claim that SAM68 tunes an endothelial morphogenetic program has been toned down in the revised manuscript. Nonetheless, our data clearly show that SAM68 is a major regulator of endothelial adhesion and conditioning of the subendothelial basement membrane.

Altogether, the proposed experiments and revisions will solidify our data and improve our study thus providing “*a significant advance towards understanding the multiple roles of RNA-binding proteins and their coordination in a study system with physiologically relevant connections*”, as stated by Reviewer#3.

Description of the planned revisions

To answer critical points raised by the 3 referees, we plan to implement our work with 3 main sets of experiments:

Set 1 of experiments: Analysis of direct interaction between SAM68 and b-actin mRNA by RIP in endothelial cells according to an improved version of published protocols and results from (Li and Richard, 2016).

Set 2 of experiments: Analysis of a direct interaction between SAM68 and the FN1 promoter by ChIP in endothelial cells according to published protocols and results from (Li and Richard, 2016).

Set 3 of experiments: Assessment of the dual functions of SAM68 and their interconnections by (i) FN rescue (expression of exogenous FN in SAM68-depleted cells) or (ii) by expression of SAM68 mutants.

In addition, we are generating tools to address the dynamic localization of b-actin in endothelial cells following SAM68 perturbations in endothelial cells (MS2 lentiviral constructs and antisense oligonucleotides designed to abrogate SAM68 recruitment onto b-actin mRNA).

Below, we describe how these sets of experiment will address Reviewers’ comments and queries in a point-by-point reply.

Our point-by-point responses regarding the revisions, including the outcomes of our proposed sets of experiments, our new results and how we have incorporated them in the manuscript and figures of the final revised version are indicated below in bold blue text.

Reviewer #1– The authors claim that SAM68 interacts with B-actin mRNA to delivery to sites of adhesion only based on siRNA-mediated knockdown experiments. Is the binding of SAM68 to B-actin dynamic process that changes with time? The authors could perform RIP experiments at different stages of cell adhesion – from early points when SAM68 is peripheric to later stages when it homogeneously distributed – to show a potential dynamic interaction with B-actin mRNA.

β-actin mRNA has been previously identified as a direct target of SAM68 in several published works performed by different groups (Itoh et al., 2002; Klein et al., 2013; Mukherjee et al., 2019). SAM68 binding site has been mapped to a 50 nt length sequence located in the 3’UTR of β-actin mRNA but direct binding of SAM68 onto β-actin mRNA has never been shown in endothelial cells. To this end, we will perform RIP experiments (Set 1 of experiments) to first identify direct recruitment of SAM68 to β-actin mRNA in endothelial cells (as suggested by reviewer 2 as well). Secondly, to address the dynamics of SAM68 interactions with β-actin mRNA we will assess direct interactions of SAM68 with β-actin mRNA at different stages of cell adhesion. These experiments will be conducted using an adapted version of the published SAM68 RIP protocol (Li and Richard, 2016).

In order to demonstrate the direct involvement of SAM68 in β-actin mRNA localization at ectopic adhesion sites in endothelial cells, we transfected cells with antisense blocking oligonucleotides that target the SAM68-binding site on b-actin mRNA and are known to abrogate SAM68 and β-actin mRNA interaction. As shown in new Figure 4C and discussed in the text of the revised manuscript, these blocking oligonucleotides significantly decreased the localization of β-actin mRNA in adhesion sites thus strengthening our claim of SAM68-mediatd delivery of mRNA to nascent adhesion sites and indicating a direct effect of SAM68 via interaction with b-actin mRNA.

Regarding the dynamics of SAM68 interaction with b-actin mRNA, RIP experiments performed at different stages of cell adhesion did not reveal any significant differences and therefore we have not included these results in our revised manuscript.

– The article would substantially benefit from live visualisation of B-actin localisation with MS2 tagged transcripts in SAM68 knockdown contexts. This would solidify the proposed mRNA delivery SAM68-mediated mechanism. Although this should not be hard to carry out given the availability of MS2-labelled animals, I understand access to the tools may constitute a major hurdle.

As mentioned by Reviewer 1, access to MS2-labelled animals and carrying out in vivo experiments in mouse endothelial cells would be a roadblock for our team in the context of this work. Nonetheless, we fully agree that live visualization of β-actin mRNA recruitment at adhesions would solidify our hypothesis. Therefore, we are currently setting up *in cellulo* experiments in endothelial cells to visualize MS2-β-actin reporters (Yoon et al., 2016), in presence of control or SAM68 binding site-directed antisense blocking oligonucleotides, as previously described (Klein et al., 2013).

Despite our attempts, it was not possible to generate stable MS2- β-actin reporter expressing primary endothelial cells to visualize live SAM68-dependent β-actin localization. However, the transfection of SAM68 binding site-directed antisense blocking oligonucleotides (new Figure 4C) has allowed us to confirm the direct implication of SAM68 in endogenous β-actin mRNA delivery to adhesion sites in endothelial cells.

Minor comment:– Could the authors run the eGFP-SAM68 movies for longer periods to show the dynamic localisation of the protein during spreading? These experiments would support the data based on fixed material.

We thank Reviewer 1 for this suggestion and will adjust our imaging pipeline for longer time acquisitions taking caution not to impact cell dynamics due to extended laser exposure.

We have performed live imaging of spreading eGFP-SAM68/LifeAct-Ruby expressing cells at longer times in an attempt to capture the dynamics of the rapidly-moving SAM68 particles. Unfortunately, and despite our attempts to adjust our imaging pipeline, the illumination of these cells for more than a few minutes induces phototoxicity (retraction of cell edges). Therefore, we are unable to include additional movies for longer periods to capture *with confidence* the dynamic localization of the protein during spreading.

Reviewer #2– The authors reference previous work defining SAM68 as a β-actin mRNA interacting protein, however, experiments confirming this in endothelial cells and that this occurs during normal focal adhesion assembly are important.

This concern will be addressed by Set 1 of experiments (RIP assays) as described in our response to the comments of Reviewer 1.

– Likewise, experiments addressing how important this action is for focal adhesion function are critical. For example, the β-actin RNA-binding site of SAM68 could be identified and perturbed to assess the direct impact of this mRNA delivery on FAK-Y397 phosphorylation, focal adhesion assembly, adhesion, cell spreading and migration/sprouting. Without these or similar experiments, the importance of SAM68-mediated β-actin mRNA delivery is unknown.

The β-actin RNA-binding site of SAM68 has previously been identified (Itoh et al., 2002) and antisense blocking oligonucleotides designed to target this sequence have been shown to abrogate SAM68 recruitment onto β-actin mRNA in neurons (Klein et al., 2013). In order to determine whether SAM68 delivery of β-actin mRNA is directly involved in focal adhesion assembly and signalling, we will use the published antisense oligonucleotides to block SAM68 recruitment and assess FAK-Y397 phosphorylation in our bead model.

– Indeed, if this is not important for FAK-Y397 phosphorylation and focal adhesion assembly, then experiments need to be designed to assess how SAM68 achieves FAK phosphorylation/maturation to provide any significant insight into SAM68 function.

To address this point, we have generated an RNA binding mutant of SAM68 (KH domain) for analysis of FAK phosphorylation using the bead assay. Importantly, SAM68 is a multi-domain protein that harbors protein/protein interaction domains (SH2 and SH3 binding domains) and it is known to act as a scaffolding protein, in TNFRα signaling for instance (Ramakrishnan and Baltimore, 2011). Therefore, we have also generated lentiviral constructs containing mutations in the SH2 or SH3 binding domains of SAM68 to interrogate its potential signaling adaptor function.

Our responses to the above comments of this Reviewer are indicated below.

Regarding the interaction of SAM68 and b-actin mRNA in endothelial cells:

As indicated in our response to Reviewer 1, we believe that our new results showing that the expression in endothelial cells of blocking antisense oligonucleotides targeting the SAM68 binding site in the 3′ UTR of β-actin mRNA decreases β-actin mRNA delivery to adhesion sites confirm that SAM68 is a b-actin mRNA interacting protein in these cells. Further, they indicate that this interaction is directly involved in β-actin mRNA localization during focal adhesion assembly (new Figure 4C).

Regarding the importance of this action (SAM68 mRNA binding) for focal adhesion function:

To interrogate how important SAM68 mRNA-binding activity is for focal adhesion signaling we generated functional mutants of the protein, depicted in new Figure 3D. Using an RNA binding-deficient mutant of SAM68 (SAM68 DKH) we first determined that RNA binding activity of SAM68 is critical for integrin signaling, assessed by FAK-Y397 phosphorylation. Further we determined that FAK-Y397 phosphorylation also involves the scaffolding activity of SAM68, using a SH3-binding mutant of SAM68 (SAM68 P358A) with impaired Src binding. These results have been included in the revised Figure 3D and detailed in the text.

– The data as presented suggest that a key function of SAM68 is to drive fibronectin (and perhaps other ECM gene) transcription. However, more experiments are needed to validate this conclusion. For example, increased FN1 promoter activity in luciferase assays may be an indirect consequence of feedback to the promoter upon SAM68-mediated action on, amongst other possible actions, focal adhesion signaling, FN transcript splicing or ECM remodeling. Experiments confirming that SAM68 interacts with the endogenous ECM gene promoter would be critical (e.g. via ChIP), as would disruption of the trans-activating action of SAM68 to directly assess the impact of this function (versus modulation of focal adhesion dynamics) on focal adhesion assembly, adhesion, cell spreading and migration/sprouting.

We fully agree that ChIP experiments to identify recruitment of SAM68 onto the endogenous FN1 promoter in endothelial cells would be required to confirm direct transcriptional activation of the *FN1* gene in these cells. Therefore, we will perform these experiments (Set 2) according to a published SAM68 ChIP protocol (to be adapted for endothelial cells) which allowed for the demonstration of specific recruitment of SAM68 onto P21 or PUMA promoters, as well as its transcriptional co-activating activity (Li and Richard, 2016).

Regarding possible indirect effects of *FN1* promoter activity in the luciferase assay shown in Figure 1F on HEK293 cells, we would like to point out that, in addition to their high transfection efficiency, HEK293 cells were chosen for this assay because they display nearly undetectable expression of FN and they are unable to assemble the molecule (even upon overexpression of exogenous FN, see Efthymiou G et al., JCS 2021). Thus, our results using this system support a direct effect of SAM68 on FN promoter activity. This information has been added to the revised text.

Results of ChIP experiments showing direct recruitment of SAM68 to the *FN1* promotor have been added to the revised version of the manuscript (revised Figure 5G and text p. 10).

– In parallel, rescue experiments to determine how recovery of endothelial FN expression impacts adhesion, cell spreading, and migration/sprouting (upon SAM68 knockdown) would determine how important this action is to control of endothelial cell behavior.

Our previous published data showed that autocrine FN expression regulates adhesion, spreading and migration of endothelial cells and that differences in FN expression levels affect assembly of the protein (Cseh et al., 2010; Radwanska et al., 2017). In SAM68-depleted cells, with compromised FN expression, the rescue of FN expression should allow us to uncouple SAM68 functions at adhesion sites from its role as a transcriptional regulator of FN expression (Set 3 of experiments). Expression of exogeneous FN in SAM68-depleted endothelial cells will be performed using lentiviral FN expression constructs described by our team (Efthymiou et al., 2021).

– Likewise, experiments designed to determine if broader disruption of COL8A1, POSTN, FBLM1 and BGN expression are direct (or indirect, e.g., due to FN disruption) would be important to understand SAM68 function.

The same set of experiments (Set 3) will be used to analyze by qRT-PCR the expression of *COL8A1, POSTN, FBLN1* and *BGN* mRNAs upon the rescue of FN expression in SAM68-depleted cells.

We were unable to stably express exogenous FN using the above-mentioned expression constructs in our primary endothelial cell cultures and thus unable to explore possible indirect effects (due to FN disruption) of SAM68 knockdown on expression of other basement components.

– Loss of SAM68 expression in other cell types is known to perturb migration, whereas migration is enhanced in endothelial cells upon SAM68 knockdown. Why would this be the case? Is it that the proposed negative impact of FN production on motility is greater than the positive impact of SAM68 focal adhesion dynamics in endothelial cells versus other cell types? Exploration of the relative impact of these proposed dual functions (using additional experiments as mentioned above) is critical to make sense of these somewhat conflicting observations.

This point relating to the balance between the *negative impact of SAM68-stimulated FN production on motility* and the *positive impact of SAM68 on focal adhesion dynamics in endothelial cells,* is very interesting. Set 3 of experiments, which includes expression of exogenous FN and assessment of cell motility in SAM68-depleted endothelial cells, should allow us to clarify this issue.

As mentioned in our response to the previous point, we were unable to stably express exogenous FN using the above-mentioned expression constructs in our primary endothelial cell cultures to assess the relative impact of the dual functions of SAM68 on cell migration. However, as indicated in the Discussion section (p. 14), we maintain that experimental conditions (different cell types and adsorbed substrates *versus* no coat) could account, at least in part, for the differences observed in other studies, compared to the present work and previous reports on endothelial and glioblastoma cells (Cseh et al., 2010; Serres et al., 2014).

Previous work has implicated phosphorylation of SAM68 as a key trigger of its activity (Locatelli and Lange, 2011, Naro et al., 2022). Additional work exploring the impact of SAM68 phosphorylation on focal adhesion dynamics and ECM gene expression/remodeling (e.g. using phospho-mutants) in this manuscript would have strengthened the message.

The regulation of SAM68 activity by phosphorylation is a complex question as SAM68 has multiple sites of phosphorylation by serine/threonine and tyrosine kinases. One of these sites (Y440) is a known substrate of Src, a major kinase activated at the cell membrane during adhesion. We are currently generating a Src phosphorylation mutant of SAM68 (Y440F) which could be used to address the impact of SAM68 phosphorylation on integrin signaling and ECM gene expression/remodeling.

As indicated in our previous response to this comment, regulation of SAM68 by tyrosine and serine/threonine phosphorylation (Locatelli and Lange, 2011, Naro et al., 2022) is complex and difficult to dissect with only one mutant, and elucidating this complex regulation is beyond the scope of the present study.

However, in light of the implication of SAM68 in FAK autophosphorylation on its Src binding site (tyrosine 397), and the importance of FAK-Src signaling in integrin signaling we generated a Src-binding-deficient mutant of SAM68. Interestingly, expression of this mutant in endothelial cells slightly increased the number of pFAK-Y397 foci around beads, as shown in new Figure 3D. These results are in line with previous studies reporting a role for interactions between SAM68 and Csk (C-terminal Src kinase/endogenous inhibitor of c-Src) in the downregulation c-Src following cell adhesion to FN. Further, they confirm a role, albeit complex, for SAM68 in the regulation of integrin signaling and focal adhesion dynamics. (See text p. 8 and 13 and for detailed discussion.)

Reviewer #3– The authors describe the observed phenotypes as resulting from 'coalescent activities' of SAM68 that play a role in the adaptation of ECs to the extracellular environment. However, it is unclear whether and which of the observed effects result from direct local functions of different SAM68 pools, versus reflecting indirect downstream consequences of one major function. For example, the effects on transcription could be a result of altered adhesion signaling and might occur independently of nuclear SAM68. Or the effects on adhesions could be an indirect consequence of altered transcription of ECM genes, independent of the transient accumulation of SAM68 at the periphery. To support that these are distinct and direct SAM68 functions, the authors would have to provide more evidence for the involvement of SAM68 in the studied processes (e.g. is SAM68 observed by CHIP at promoter regions of ECM genes whose transcription is affected?)

As recommended by Reviewer 2 as well, we will perform ChIP experiments to document the direct recruitment of SAM68 onto the *FN1* promoter (Set 2 of experiments).

As indicated in the response to Reviewer 2, the ChIP experiments have been performed and included in Figure 5G

– Try to uncouple them to assess their relative contributions and potential connections in the observed phenotypes e.g. it would be informative to attempt to rescue the knockdown phenotypes with mutants of SAM68 that cannot be imported into the nucleus or that cannot bind RNA or that cannot be phosphorylated by Src

Set 3 of experiments should allow us to uncouple dual functions of SAM68 in endothelial cells. In these experiments, integrin signaling defects will be evaluated in SAM68–depleted cells following the rescue of FN expression. Persistence of the adhesion site defect would indicate that transcriptional activity and adhesion site regulation by SAM68 are distinct events. Moreover, as indicated above, we are generating lentiviral constructs of SAM68 mutants with impaired ability to bind RNA or be phosphorylated by Src (Y440F), in order to assess their effect on integrin signaling.

As indicated in the response to Reviewer 2, we were unable to successfully carry out FN rescue experiments proposed in set 3 of experiments. However, as mentioned in our response to Reviewer 2, we were able to examine the functional consequences of rescuing SAM68 knockdown with mutants deficient in RNA binding/processing or impaired for Src kinase binding/signaling. We believe that the results of these experiments (new Figure 3D), together with results of the ChIP experiments (new Figure 5G) described in the revised manuscript support distinct and direct functions of SAM68.

Minor comment:Also, is the effect of SAM68 depletion on pY397-FAK levels local and/or transient? it would be useful to present data on the total amount of pY397-FAK (by IF or western) in control and si-SAM68 cells at early and late stages of spreading

This point is very interesting and will be tested at early vs late stage of spreading.

Due to time constraints and prioritization, we were not able to undertake these longitudinal analyses.

Description of the revisions that have already been incorporated in the transferred manuscriptReviewer#1– The authors state that "…both submembranous functions […] and nuclear functions […] of SAM68 contribute to the morphogenetic phenotype of angiogenic endothelial cells. Some caution must be taken, as all previous data were obtained from 2D experiments. At this stage it cannot be excluded other mechanisms involved in 3D migration.

We fully agree with this reviewer’s comment and we have modified the manuscript to take into account the fact that we cannot exclude other mechanisms of action for SAM68 in 3D endothelial cell sprouting experiments However, it is noteworthy that the migration per se of individual cells is not measured in our 3D experiments.

Minor comments:– In Figure 1F, there is a drop-in luciferase activity in cells transfected with higher amounts of vector, rather than an increase with SAM68. Why?

The luciferase reporter assay is a convenient and well-accepted means of evaluating promoter activities, however, it requires the transfection of increasing amounts of expression plasmids, which often contain strong promoters such as CMV (in our case). Depending on the experimental conditions, a drop in luciferase activity is often observed, due to titration of general transcriptional factors. In our experiments shown in Figure 1F, despite the observed drop in luciferase activity in pcDNA 3.1-transfected cells, transfection of increasing amounts of the SAM68 expression vector induced a significant increase in luciferase activity.

– The authors claim (and rightly so) that plasmids are hard to transfect into HUVECs when describing luciferase reporter assays. However, they express eGFP-SAM68 (presumably from a plasmid).

eGFP-SAM68 was delivered and expressed in endothelial cells using a lentiviral vector (this has been specified in the revised manuscript: legend to Movie supplement 1). Although eGFP-SAM68 is successfully expressed, the efficiency of infection is a bit low. Thus, this method is adequate when experiments require observations at the single cell level, such as imaging of endothelial cells expressing eGFP-Sam68. However, the low infection efficiency makes it unsuited for the observation of global effects on the cell population, as is the case for a luciferase assay in which all cells from a given experimental condition are lysed.

– Some experimental details in the figure legends could be restricted / moved to the methods section.– Some typos and British/American spelling inconsistencies (e.g. localisation and localization) need to be corrected throughout the manuscript.– Statistical analysis details could be mentioned in figure legends.– In page 11, "… proposed to be involved in regulation of early cell adhesion processes and spreading" needs referencing.– Y axes in some graphs do not start at 0, which may mislead visual interpretations.– "Figure 2—figure supplement 1" in page should read "movie supplement 1"

We thank Reviewer #1 for these comments which have all been taken into account. Appropriate changes/corrections have been made in the revised version of the manuscript.

Reviewer#2– The title of the manuscript states that SAM68 modulates the morphogenetic program of endothelial cells, yet there are no studies of blood vessel morphogenesis described by this work. Ultimately, in vivo studies of vessel development in SAM68 mutant mice would be required to be able to make this claim.

We agree that only endothelial cell morphogenesis, and not blood vessel morphogenesis, has been addressed in this study. In light of the reviewer’s recommendation to tone down claims that SAM68 tunes an endothelial morphogenetic program, we have modified the revised manuscript text and title.

– Place the work in the context of the existing literature (provide references, where appropriate).SAM68 has previously been identified as an RNA-binding protein associated with the 'adhesome' that regulates cell motility (Huot et al., 2009a, Locatelli and Lange, 2011, Naro et al., 2022). Here, Rekad and colleagues also probe the action of SAM68 in endothelial cell migration, but find this to be enhanced upon SAM68 knockdown – unlike previous studies demonstrating a reduction in motility in similar experiments in other cell types. Indeed, a detailed discussion of this discrepancy would have been appreciated.

As recommended by Reviewer #2, we have included a more detailed discussion of this point in the revised manuscript.

Reviewer #3Figure 3C: Is the n=3 indicative that only 3 beads were analyzed? Given the relatively small difference, a larger sample size would be useful.

We thank the Reviewer for pointing out this mistake. Three independent experiments have been performed with quantification of at least 12 beads for each condition. The manuscript has been corrected accordingly (N=3).

Page 5-6: The statement 'nearly all adhesion sites in SAM68-depleted cells remained smaller than 0.75 um' doesn't seem to accurately reflect the data presented in the right panel of Figure 1C.

We have modified the units (µm^2^) of average adhesion size. Nearly all adhesion sites in SAM68-depleted cells remained smaller than 0.75 µm^2^

Page 6: there is a reference to a G418 phosphotyrosine antibody. Do the authors mean 4G10 antibody? Also, there is a mention that materials are listed in Supplemental tables 1 and 2, but these were not attached.

We thank the Reviewer for having noted these typos, and the fact that we omitted to attached Supplemental Tables 1 and 2. This has been corrected in the revised manuscript, to be submitted with the Supplemental Tables.

Description of analyses that authors prefer not to carry outReviewer #1– deHoog et al. 2004 (doi.org/10.1016/S0092-8674(04)00456-8) had shown the presence of SAM68 in SICs. Why do the authors believe that the presence of SAM68 in the periphery in endothelial cells does not mark the formation of SICs in these cells?

Spreading Initiation Centers (SICs) are described as structures involved in the early step of adhesion which contain SAM68, along with other RNA binding proteins (de Hoog et al., 2004) in MRC5 cells. In the same paper, to test whether SICs are a general feature of cell adhesion, authors evaluated the presence of SICS during adhesion of several other cell including endothelial cells (HUVEC). Among the 6 types of cells tested, SICs were not observed in nonfibroblastic cell types. In accordance with this study, we did not observe SICs, as defined by deHoog et al., in endothelial cells plated onto FN

– In Shestakova et al. 2001 (doi.org/10.1073/pnas.121146098), decreased localisation of B-actin mRNA leads to reduced persistence of direction of movement. Was this measured? Is this not seen here because SAM68 is only responsible for B-actin mRNA localisation at early stages of adhesion?

We thank Reviewer 1 for this comment. After re-analysis of our migration data we did not detect a significant effect on the persistence of migration in our experimental conditions. This could indeed reflect the temporal regulation by SAM68 of b-actin mRNA localization at the leading edge of cells, although we cannot exclude additional defects caused by SAM68 depletion on adhesion stability and lammelipodial protrusion and consequently cell polarity and directional motility.

– Although the authors claim that altered ECM deposition in SAM68 deficient cells results from altered transcription, they do not address potential misregulation of translation and secretion.

We did not address misregulated translation here as FN mRNA levels were significantly decreased in SAM68-depleted cells. The decreased transcript levels were accompanied by decreased protein levels. Upon depletion of SAM68, we detected less FN in both “soluble” (conditioned medium) and “insoluble” (ECM-associated) forms, as shown in the western blots of Figure 4—figure supplement 1. We do not believe that SAM68 silencing impacts FN secretion, as we did not observe differential retention of FN in the cytoplasm of SAM68-depleted cells compared to control cells by immunostaining (Figure 4C). Rather, FN staining was strictly fibrillar (ECM-associated) in both control and SAM68-depleted cells, and the intensity profile baseline values were similarly low. This point has been added to the revised manuscript.

– In fact the highlight that whilst the level of some mRNAs encoding basement membrane proteins do not decrease in the absence of SAM68, their incorporation was severely affected. This is worth exploring to strengthen the manuscript.

This issue was not addressed for other basement membrane components. However, the dichotomy in expression and matrix incorporation of certain basement membrane components is most likely due to the sequential and hierarchical nature of ECM assembly. FN is one of the earliest ECM proteins to be assembled and observations from multiple laboratories have shown that FN orchestrates the assembly of multiple matrix components (reviewed in, (Dallas et al., 2006; Marchand et al., 2019)), including COLIV ((Filla et al., 2017; Miller et al., 2014)).

– Whilst the data presented in figure 7 is convincing, some more detailed mechanistic analyses could help further comprehend 2D and 3D behaviours. Could it be that the nuclear and cellular roles of SAM68 are somewhat decoupled depending on the environment? Could the RNA localisation functions have a critical role in endothelial sprouting and not so much in 2D migration? Some insights are needed to address these questions and wrap up some loose ends. In its current form, this section of the manuscript is too vague.

It is known that 2D and 3D culture conditions induce differences in cell behavior, notably through differences in the physical (rigid vs pliable) and biochemical (plastic vs fibrin gel) nature of the environments that differentially regulate mechanotransduction, integrin signaling, cell polarity, etc. Here, we show that migration of endothelial cells on rigid 2D substrates is increased upon SAM68 depletion. On the other hand, the ability of cells to align in capillary-like cords and invade a 3D environment is reduced. Mechanistically, effects of SAM68 on FN production and ECM assembly are likely involved in both contexts by providing an adhesive substrate that restricts cell motility in 2D, or bridges neighboring cells and promotes cell survival in 3D. The purpose of performing the sprouting assay presented here in addition to cell migration assays was not to compare the same functions of SAM68 in these 2 different contexts but rather to illustrate that SAM68 controls endothelial cell behavior in both 2D and 3D environments and thus could significantly impact angiogenesis.

Reviewer#2– Many of the findings are rather superficial or observational, and a detailed mechanistic understanding of SAM68 function is lacking. For example, loss of SAM68 expression reduces β-actin mRNA recruitment to sites of fibronectin-coated bead adhesion, but how is this regulated and what is its impact on focal adhesion dynamics?

Both the role of β-actin mRNA localization on cell adhesion dynamics and the impact of reducing this localization have been extensively documented (Katz et al., 2012; Kislauskis et al., 1994; Shestakova et al., 2001), or (Herbert and Costa, 2019) and references therein. In particular, a specific RNA binding protein called ZBP1 has been shown to localize actin mRNA near focal adhesions (Katz et al., 2012) by a Src kinase-dependent mechanism (Hüttelmaier et al., 2005).

References

Cseh B, Fernandez-Sauze S, Grall D, Schaub S, Doma E, Van Obberghen-Schilling E. 2010. Autocrine fibronectin directs matrix assembly and crosstalk between cell-matrix and cell-cell adhesion in vascular endothelial cells. *J Cell Sci* 123:3989–3999. doi:10.1242/jcs.073346

Dallas SL, Chen Q, Sivakumar P. 2006. Dynamics of Assembly and Reorganization of Extracellular Matrix ProteinsCurrent Topics in Developmental Biology. Academic Press. pp. 1–24. doi:10.1016/S0070-2153(06)75001-3

de Hoog CL, Foster LJ, Mann M. 2004. RNA and RNA binding proteins participate in early stages of cell spreading through spreading initiation centers. *Cell* 117:649–662. doi:10.1016/s0092-8674(04)00456-8

Efthymiou G, Radwanska A, Grapa A-I, Beghelli-de la Forest Divonne S, Grall D, Schaub S, Hattab M, Pisano S, Poet M, Pisani DF, Counillon L, Descombes X, Blanc-Féraud L, Van Obberghen-Schilling E. 2021. Fibronectin Extra Domains tune cellular responses and confer topographically distinct features to fibril networks. *J Cell Sci* 134:jcs252957. doi:10.1242/jcs.252957

Filla MS, Dimeo KD, Tong T, Peters DM. 2017. Disruption of fibronectin matrix affects type IV collagen, fibrillin and laminin deposition into extracellular matrix of human trabecular meshwork (HTM) cells. *Exp Eye Res* 165:7–19. doi:10.1016/j.exer.2017.08.017

Herbert SP, Costa G. 2019. Sending messages in moving cells: mRNA localization and the regulation of cell migration. *Essays Biochem* 63:595–606. doi:10.1042/EBC20190009

Hüttelmaier S, Zenklusen D, Lederer M, Dictenberg J, Lorenz M, Meng X, Bassell GJ, Condeelis J, Singer RH. 2005. Spatial regulation of β-actin translation by Src-dependent phosphorylation of ZBP1. *Nature* 438:512–515. doi:10.1038/nature04115

Itoh M, Haga I, Li Q-H, Fujisawa J. 2002. Identification of cellular mRNA targets for RNA-binding protein Sam68. *Nucleic Acids Res* 30:5452–5464. doi:10.1093/nar/gkf673

Katz ZB, Wells AL, Park HY, Wu B, Shenoy SM, Singer RH. 2012. β-Actin mRNA compartmentalization enhances focal adhesion stability and directs cell migration. *Genes Dev* 26:1885–1890. doi:10.1101/gad.190413.112

Kislauskis EH, Zhu X, Singer RH. 1994. Sequences responsible for intracellular localization of β-actin messenger RNA also affect cell phenotype. *J Cell Biol* 127:441–451. doi:10.1083/jcb.127.2.441

Klein ME, Younts TJ, Castillo PE, Jordan BA. 2013. RNA-binding protein Sam68 controls synapse number and local β-actin mRNA metabolism in dendrites. *Proc Natl Acad Sci U S A* 110:3125–3130. doi:10.1073/pnas.1209811110

Li N, Richard S. 2016. Sam68 functions as a transcriptional coactivator of the p53 tumor suppressor. *Nucleic Acids Res* 44:8726–8741. doi:10.1093/nar/gkw582

Marchand M, Monnot C, Muller L, Germain S. 2019. Extracellular matrix scaffolding in angiogenesis and capillary homeostasis. *Semin Cell Dev Biol*, Mammalian innate immunity to fungal infection 89:147–156. doi:10.1016/j.semcdb.2018.08.007

Miller CG, Pozzi A, Zent R, Schwarzbauer JE. 2014. Effects of high glucose on integrin activity and fibronectin matrix assembly by mesangial cells. *Mol Biol Cell* 25:2342–2350. doi:10.1091/mbc.e14-03-0800

Mukherjee J, Hermesh O, Eliscovich C, Nalpas N, Franz-Wachtel M, Maček B, Jansen R-P. 2019. β-Actin mRNA interactome mapping by proximity biotinylation. *Proc Natl Acad Sci* 116:12863–12872. doi:10.1073/pnas.1820737116

Radwanska A, Grall D, Schaub S, Divonne SB la F, Ciais D, Rekima S, Rupp T, Sudaka A, Orend G, Van Obberghen-Schilling E. 2017. Counterbalancing anti-adhesive effects of Tenascin-C through fibronectin expression in endothelial cells. *Sci Rep* 7:12762. doi:10.1038/s41598-017-13008-9

Ramakrishnan P, Baltimore D. 2011. Sam68 Is Required for Both NF-κB Activation and Apoptosis Signaling by the TNF Receptor. *Mol Cell* 43:167–179. doi:10.1016/j.molcel.2011.05.007

Shestakova EA, Singer RH, Condeelis J. 2001. The physiological significance of β -actin mRNA localization in determining cell polarity and directional motility. *Proc Natl Acad Sci U S A* 98:7045–7050. doi:10.1073/pnas.121146098

Yoon YJ, Wu B, Buxbaum AR, Das S, Tsai A, English BP, Grimm JB, Lavis LD, Singer RH. 2016. Glutamate-induced RNA localization and translation in neurons. *Proc Natl Acad Sci U S A* 113:E6877–E6886. doi:10.1073/pnas.1614267113